# Adaptation Accelerating Sampling-based Bayesian Inference in Attractor Neural Networks

**Xingsi Dong**[1]
dxs19980605@pku.edu.cn

**Zilong Ji**[1,3]
zilong.ji@ucl.ac.uk

**Tianhao Chu**[1]
chutianhao@stu.pku.edu.cn

**Tiejun Huang**[4]
tjhuang@pku.edu.cn

**Wen-Hao Zhang**[2,†]
wenhao.zhang@utsouthwestern.edu

**Si Wu**[1,†]
siwu@pku.edu.cn

1, School of Psychology and Cognitive Sciences, IDG/McGovern Institute for Brain Research,
PKU-Tsinghua Center for Life Sciences, Academy for Advanced Interdisciplinary Studies,
Center of Quantitative Biology, Peking University.
2. Lyda Hill Department of Bioinformatics, O'Donnell Brain Institute, UT Southwestern Medical Center.
3. Institute of Cognitive Neuroscience, University College London
4. School of Computer Science, Peking University.
[†]: Corresponding authors.

## Abstract

The brain performs probabilistic Bayesian inference to interpret the external world. The sampling-based view assumes that the brain represents the stimulus posterior distribution via samples of stochastic neuronal responses. Although the idea of sampling-based inference is appealing, it faces a critical challenge of whether stochastic sampling is fast enough to match the rapid computation of the brain. In this study, we explore how latent feature sampling can be accelerated in neural circuits. Specifically, we consider a canonical neural circuit model called continuous attractor neural networks (CANNs) and investigate how sampling-based inference of latent continuous variables is accelerated in CANNs. Intriguingly, we find that by including noisy adaptation in the neuronal dynamics, the CANN is able to speed up the sampling process significantly. We theoretically derive that the CANN with noisy adaptation implements the efficient sampling method called Hamiltonian dynamics with friction, where noisy adaption effectively plays the role of momentum. We theoretically analyze the sampling performances of the network and derive the condition when the acceleration has the maximum effect. Simulation results validate our theoretical analyses. We further extend the model to coupled CANNs and demonstrate that noisy adaptation accelerates the sampling of the posterior distribution of multivariate stimuli. We hope that this study enhances our understanding of how Bayesian inference is realized in the brain.

## 1 Introduction

A large volume of human behavioral [1–3] and animal neurophysiological studies [4, 5] have suggested that the brain performs statistically optimal Bayesian inference to interpret the external world [6–8]. Yet, exactly how neural circuits in the brain implement probabilistic inference (algorithm) and how neuronal responses represent the stimulus posterior distribution (representation) remain debated. Among the proposed models in the literature [9–15], the sampling-based view is promising [6, 12–19], which considers that the stimulus posterior distribution is approximately represented by samples generated by the neural system over time, with each sample coming from the

stochastic neuronal responses. The view of stochastic sampling naturally accounts for the irregular firing and other response properties of neurons observed in the experiments [17, 18, 20].

Although the idea of sampling-based inference is appealing, a critical concern is whether stochastic sampling is fast enough to match the rapid computation of the brain. For instance, the sampling trajectory of Gibbs sampling [21, 22] or Langevin sampling [23] essentially performs random walks in local regions rather than the whole posterior space, which is too slow to be compatible with brain functions [15, 24]. Thus, it is crucial to explore whether neural circuits in the brain have the capacity of realizing sampling-based inference rapidly. This issue has been investigated recently by several works [15, 16, 18, 25]. Among them, a computational model which considers that neural circuits are performing Hamiltonian Monte Carlo (HMC) sampling is attracting [16, 18]. HMC is a method developed in machine learning for accelerating stochastic sampling [26]. In HMC, an auxiliary variable representing the momentum of the sampled space variable is introduced, and the two variables follow the Hamiltonian dynamics. By sampling the momentum variable stochastically from a simple normal distribution, it achieves that the sampling speed of the space variable is accelerated significantly compared to that of random walks. Interestingly, the study [16] found that HMC can be implemented by a biologically plausible neural network with balanced excitation and inhibition (E-I) interactions among neurons, where the activities of inhibitory neurons effectively serve as the auxiliary 'momentum' to accelerate the sampling of responses of excitatory neurons.

Previous works have mainly studied the case that the sampled features are represented by individual neurons [15–18, 20]. In the brain, however, it is known that some continuous variables are encoded jointly by a population of neurons called population coding [27]. The well-known examples include orientation [28], moving direction [29], head-direction [30], and spatial location [31, 32], and the representations of these continuous features are typically mimicked by a canonical network model called continuous attractor neural networks (CANNs). Thus, in this study, we investigate how sampling-based inference of a continuous feature in a CANN is accelerated. In the brain, there is an added difficulty as the brain typically extracts and represents multiple features in a parallel and distributed fashion using separated neural circuits, e.g., for multisensory integration [5, 33] and contour integration [34, 35], indicating that the acceleration of distributed sampling-based inference is also important. Thus, we also investigate how distributed sampling-based inference is accelerated in coupled CANNs.

In this work, we find that by including noisy adaptation, a generic feature of neuronal responses, in the dynamics of a CANN, the network is able to speed up sampling-based inference significantly. The underlying mechanism can be intuitively understood as follows. In a CANN, the values of a continuous feature are encoded by a continuous family of localized stationary states (attractors) called bumps, which form a low-dimensional manifold (the attractor space) to represent stimulus feature values. Without adaptation, internal noises in the network drive the bump to exhibit Brownian motion on the attractor space, implying that the CANN performs Langevin sampling. When noisy adaptation is included in the CANN dynamics, it tends to destabilize the network bump response and causes the bump to experience history-dependent, large-step stochastic movements in the attractor space, which accelerates the sampling process effectively. Remarkably, we show that the sampling process of the CANN with noisy adaptation is equivalent to Hamiltonian dynamics with friction (HDF) [36], where the adaptation effectively serves as the 'momentum' for speeding up the sampling of bump positions (i.e., feature values). We theoretically analyze how noisy adaptation accelerates the sampling performance of the network and derive the condition when the acceleration has the maximum effect. The theoretical analyses are validated by simulations. Furthermore, we extend the study to the case of coupled CANNs, where the correlation priors between features are stored in the reciprocal connections between CANNs. We demonstrate that such coupled neural circuits can achieve distributed sampling-based inference efficiently, and noisy adaptation speeds up the sampling process in the high-dimensional feature space. We hope that this study enhances our understanding of the implementation of sampling-based inference in the brain.

## 2   A model of Bayesian inference

To study the acceleration of sampling-based Bayesian inference in neural systems, we consider a linear Gaussian generative model (Fig.1A), which has been widely used in previous studies [11, 15, 25, 37, 38]. The model assumes that an observation $s^o$ is generated by a latent feature $s$ according to

a Gaussian distribution, i.e.,

$$p(s^{\mathrm{o}}|s) = \mathcal{N}(s^{\mathrm{o}}|s, \Lambda^{-1}), \tag{1}$$

where $\mathcal{N}(s^{\mathrm{o}}|s, \Lambda^{-1})$ denotes the Gaussian distribution with mean $s$ and precision $\Lambda$ (the inverse of variance). Here, the feature can be any attribute extracted by neural systems, such as orientation, moving direction, head direction, or spatial location.

Bayesian inference computes the posterior of the latent feature $s$ according to the Bayes' theorem,

$$p(s|s^{\mathrm{o}}) \propto p(s^{\mathrm{o}}|s)p(s) = \mathcal{N}(s|s^{\mathrm{o}}, \Lambda^{-1}), \tag{2}$$

where $p(s)$ denotes the prior of the latent feature which is assumed to be uniform in this study.

When studying distributed sampling-based inference, we consider that observations and the associated latent features are high-dimensional, denoted, respectively, as $\mathbf{s}^{\mathrm{o}} = \{s_i^{\mathrm{o}}\}$ and $\mathbf{s} = \{s_i\}$, for $i = 1, \ldots, M$, with $M > 1$ the dimension of features (Fig.3A). The likelihood function and the prior of the high-dimensional Gaussian generative model are written as,

$$p(\mathbf{s}^{\mathrm{o}}|\mathbf{s}) = \mathcal{N}(\mathbf{s}^{\mathrm{o}}|\mathbf{s}, \boldsymbol{\Lambda}^{-1}), \quad p(\mathbf{s}) \propto \mathcal{N}(\mathbf{s}|\mathbf{0}, \mathbf{L}^{-1}), \tag{3}$$

where the precision matrix $\boldsymbol{\Lambda}$ of the likelihood function is diagonal, i.e., $\boldsymbol{\Lambda} = \mathrm{diag}(\Lambda_1, \Lambda_2, \cdots, \Lambda_M)$, implying that each observed feature $s_i^{\mathrm{o}}$ is independently generated by $s_i$, satisfying $p(\mathbf{s}^{\mathrm{o}}|\mathbf{s}) = \prod_{i=1}^{M} p(s_i^{\mathrm{o}}|s_i) = \prod_{i=1}^{M} \mathcal{N}(s_i^{\mathrm{o}}|s_i, \Lambda_i^{-1})$. This assumption is reasonable, since the observations of different features are through separate neural pathways. The precision matrix $\mathbf{L}$ of the prior is Laplacian, i.e., $L_{ii} = -\sum_j L_{ij}$, for $j \neq i$. This gives a uniform marginal prior for each feature and a co-occurrence probability between any two features $p(s_i, s_j) \propto \exp\left[-L_{ij}(s_i - s_j)^2/2\right]$. The Laplacian prior was used in studying contour integration and multi-sensory integration [34, 39–41]).

According to the Bayes' theorem, the posterior distribution of the latent features in the high-dimensional case is calculated to be,

$$p(\mathbf{s}|\mathbf{s}^{\mathrm{o}}) \propto p(\mathbf{s}^{\mathrm{o}}|\mathbf{s})p(\mathbf{s}) = \mathcal{N}(\mathbf{s}|\boldsymbol{\mu_s}, \boldsymbol{\Omega}^{-1}), \tag{4}$$

where the mean $\boldsymbol{\mu_s}$ and the precision matrix $\boldsymbol{\Omega}$ are given by,

$$\boldsymbol{\Omega} = \boldsymbol{\Lambda} + \mathbf{L}, \quad \boldsymbol{\mu_s} = \boldsymbol{\Omega}^{-1}\boldsymbol{\Lambda}\mathbf{s}^{\mathrm{o}}. \tag{5}$$

## 3 Hamiltonian dynamics with friction

We first review a machine learning method for accelerating sampling-based Bayesian inference, and later we embed it into a concrete neural circuit dynamics.

The first-order Langevin dynamics (FLD) has been applied to sample the posterior distribution of Bayesian inference [23, 42], and it samples stimulus values by performing stochastic gradient ascent on the manifold of the log-posterior of the stimuli, i.e.,

$$\tau_s \frac{\mathrm{d}\mathbf{s}}{\mathrm{d}t} = \boldsymbol{\alpha}\nabla \ln p(\mathbf{s}|\mathbf{s}^{\mathrm{o}}) + \sqrt{\tau_s}\boldsymbol{\sigma}_s\boldsymbol{\xi}, \tag{6}$$

where $\tau_s$ is the time constant of sampling. $\nabla = \mathrm{d}/\mathrm{d}\mathbf{s}$ denotes the derivative over $\mathbf{s}$. $\boldsymbol{\xi}$ are multivariate independent Gaussian-white noises, satisfying $\langle\boldsymbol{\xi}(t)\boldsymbol{\xi}(t')^{\top}\rangle = \mathbf{I}\delta(t - t')$, with $\mathbf{I}$ the identity matrix and $\delta(t - t')$ the Dirac delta function. $\boldsymbol{\sigma}_s$ are noise strengths, and $\boldsymbol{\alpha} = \boldsymbol{\sigma}_s\boldsymbol{\sigma}_s^T/2$.

It has been proved that the stationary distribution of FLD equals to the target posterior distribution $p(\mathbf{s}|\mathbf{s}^{\mathrm{o}})$ [23]. Under the drive of Gaussian-white noises, the value of $\mathbf{s}$ fluctuates over time, which can be regarded as samples from the posterior $p(\mathbf{s}|\mathbf{s}^{\mathrm{o}})$. FLD essentially performs noisy gradient ascent in the space of log posterior (Fig. 1C), which converges slowly. To speed up the sampling process, Hamiltonian dynamics with friction (HDF) (second-order Langevin dynamics) was proposed [36]. This approach induces a set of auxiliary momentum variables $\mathbf{y}$ to the 'space' variables $\mathbf{s}$ and meanwhile includes friction of momentum to reduce large fluctuations. The dynamics of HDF is written as,

$$\tau_s \frac{\mathrm{d}\mathbf{s}}{\mathrm{d}t} = \boldsymbol{\alpha}^{-1}\mathbf{y}, \tag{7}$$

$$\tau_z \frac{\mathrm{d}\mathbf{y}}{\mathrm{d}t} = -\boldsymbol{\beta}\boldsymbol{\alpha}^{-1}\mathbf{y} + \nabla \ln p(\mathbf{s}|\mathbf{s}^{\mathrm{o}}) + \sqrt{\tau_z}\boldsymbol{\sigma}_y\boldsymbol{\xi}, \tag{8}$$

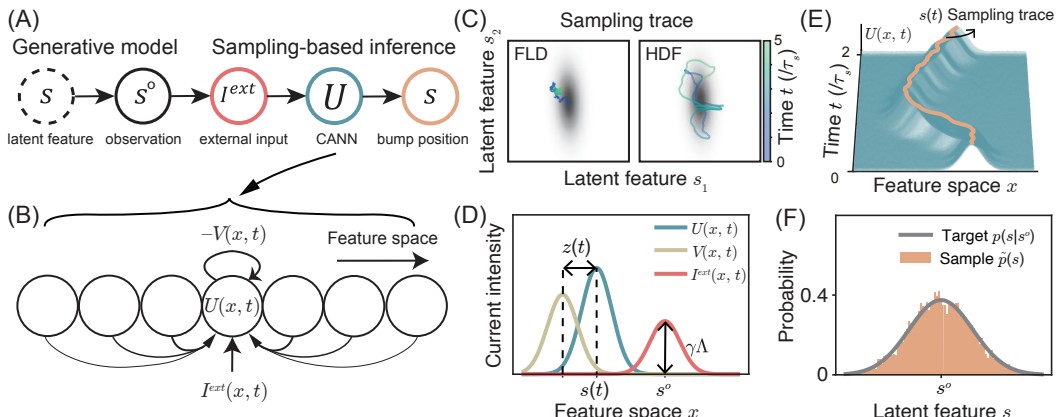

Figure 1: A CANN with noisy adaptation implements sampling-based Bayesian inference. (A) Illustration of the inference process. Generative model: a latent feature $s$ sampled from the prior $p(s)$ generates an observation $s^o$ according to the likelihood function $p(s^o|s)$. Sampling-based inference: a CANN receives the external input $I^{ext}$ conveying the likelihood function of the observation and samples the feature value $s(t)$ by the bump position. (B) The CANN structure. Neurons are uniformly distributed in the feature space and connected recurrently. Each neuron receives an external input $I^{ext}(x, t)$ and an adaptation current $-V(x, t)$. (C) Illustrating the slow sampling process of FLD and the fast sampling process of HDF. Two-dimensional case is shown. (D) The network state. The neural bump $U(x, t)$ locates at $s(t)$, i.e., the feature value represented by the network. The adaptation current $V(x, t)$ is delayed to $U(x, t)$, with a separation $z(t)$. The external input $I^{ext}(x, t)$ locates at the observation $s^o$. (E) A sampling trace of the CANN bump position (orange line) which samples feature values. (F) The sampled distribution $\tilde{p}(s)$ (the normalized orange histogram) agrees with the target posterior $p(s|s^o)$ (grey line). See SI.1 for parameter setting and simulation details.

where $\tau_s$ and $\tau_z$ are the time constants of the space variables $\mathbf{s}$ and the momentum variables $\mathbf{y}$, respectively. The matrix $\boldsymbol{\alpha}$ corresponds to the inertia of the Hamiltonian dynamics, and the term $-\boldsymbol{\beta}\boldsymbol{\alpha}^{-1}\mathbf{y}$ on the right-hand side of Eq.(8) represents the friction of momentum, with the matrix $\boldsymbol{\beta}$ controlling the friction strength. The noise strength matrix satisfies $\boldsymbol{\sigma}_y\boldsymbol{\sigma}_y^\top = 2\boldsymbol{\beta}\tau_s/\tau_z$.

In the case that the feature $s$ is one-dimensional and its posterior distribution satisfies Eq.(2), HDF is written as,

$$\tau_s \frac{\mathrm{d}s}{\mathrm{d}t} = \frac{1}{\alpha}y, \tag{9}$$

$$\tau_z \frac{\mathrm{d}y}{\mathrm{d}t} = -\frac{\beta}{\alpha}y + \Lambda(s^o - s) + \sqrt{\tau_z}\sigma_y\xi. \tag{10}$$

As illustrated in Fig. 1C, FLD wanders locally in the posterior space, while HDF travels distantly and hence speeds up the sampling.

## 4 A CANN with noisy adaptation accelerating sampling-based inference

We explore how a CANN with noisy adaptation accelerates the sampling-based Bayesian inference and demonstrate that it implements HDF.

### 4.1 A CANN with noisy adaptation

We consider that an one-dimensional continuous feature $s \in \mathbb{R}$ is encoded by a CANN, in which neurons are uniformly distributed in the feature space of $x$ (Fig.1B). Denote $U(x, t)$ as the synaptic input received by neurons at $x$, and $r(x, t)$ the corresponding firing rate. The dynamics of the network

is written as,

$$\tau_s \frac{\partial U(x,t)}{\partial t} = -U(x,t) + \rho \int_{x'} W(x,x')r(x',t)\mathrm{d}x' - V(x,t) + I^{ext}(x,t), \tag{11}$$

$$r(x,t) = \frac{U^2(x,t)}{1 + k\rho \int_{x'} U^2(x',t)\mathrm{d}x'}, \tag{12}$$

where $\tau_s$ is the synaptic time constant and $\rho$ the neuronal density. The neuronal connection strengths $W(x,x') = J_0/(\sqrt{2\pi}a) \exp\left[-(x-x')^2/(2a^2)\right]$ are translation-invariant in the space, with $a$ and $J_0$ controlling the connection range and amplitude, respectively. $I^{ext}(x,t)$ represents the feedforward input from other areas, e.g., the sensory input. The neuronal firing rate $r(x,t)$ is a nonlinear function of synaptic inputs, where the parameter $k$ in the denominator of Eq.(12) controls the amplitude of divisive normalization, which could be implemented by shunting inhibition via inhibitory neurons [43, 44].

The current $-V(x,t)$ on the right-hand side of Eq.(11) reflects the adaptation effect. Adaptation is a general phenomenon referring to that a neuron system exploits negative feedback to suppress neuronal responses when they are high, and it may arise from different mechanisms. Here we exemplify it with spike frequency adaptation [45]. The dynamics of $V(x,t)$ is written as,

$$\tau_z \frac{\partial V(x,t)}{\partial t} = -V(x,t) + mU(x,t) + \sigma_V\sqrt{\tau_z U(x,t)}\xi(x,t), \tag{13}$$

where $\tau_z$ is the time constant of adaptation and $m$ the adaptation strength. $\xi(x,t)$ denotes Gaussian white noise of zero mean and unit variance, and the parameter $\sigma_V$ controls the noise amplitude. Notably, we include noises in the adaptation dynamics, which is biologically reasonable, as noises are ubiquitous in neural systems (more discussion in Sec.6).

We review some fundamental properties of CANNs relevant to the present study. When there is no external input ($I^{ext} = 0$) or adaptation ($m = 0$, $\sigma_V = 0$), the CANN holds a continuous family of Gaussian-shaped stationary states called bumps when $k < \rho J_0^2/(8\sqrt{2\pi}a)$. These bump states can be approximately expressed as $\overline{U}(x) = A_U \exp\left[-(x-s)^2/(4a^2)\right]$, where $A_U$ denotes the bump height, and the bump position (center) $s$ denotes the feature value represented by the CANN. These bumps form an attractor space for $s \in \mathbb{R}$, on which the network is neutrally stable [46, 47]. Under the drive of Gaussian white noises, the bump position exhibits Brownian motion in the attractor space [48]. When adaptation is induced in a CANN, it destabilizes the active bump state. In particular, when the adaptation strength is larger than a boundary, i.e., $m > \tau_s/\tau_z$, the network can hold a spontaneously moving bump state called travelling wave [49, 50]. When noises are included in the adaptation and the mean of the adaptation strength is close to the travelling wave boundary $\tau_s/\tau_z$, the CANN exhibits Lévy flights [51].

In this study, we are interested in the dynamics of a CANN with noisy adaptation, when an external input conveying the stimulus information is presented. Specifically, we consider that the external input $I^{ext}(x,t) = \gamma I(x)$, with a constant $\gamma$ controlling the input strength, and $I(x)$ has the form,

$$I(x) = \Lambda \exp\left[-\frac{(x-s^o)^2}{4a^2}\right], \tag{14}$$

where $s^o$ is the observed feature and $\Lambda$ the prevision of the likelihood function given in Eq.(1). Here, we take the view of probabilistic population coding (PPC) which assumes that the mean and uncertainty of the feature are encoded by the joint activities of an ensemble of neurons [9], and we consider that the activities of these neurons provide the feedforward sensory input to the CANN.

## 4.2 The network dynamics implementing HDF

We derive that the network dynamics Eqs.(11-14) implement HDF. As a general property of CANNs, when the external input is sufficiently small (i.e., $\gamma$ is small) and that the adaptation strength $m$ is smaller than a threshold (the value of the threshold will be given later), the network state can be approximated to be of the Gaussian form (Fig.1D), which are written as,

$$U(x,t) = u_0 \mathcal{G}\left[x|s(t), 2a^2\right], \quad r(x,t) = r_0 \mathcal{G}\left[x|s(t), a^2\right], \quad V(x,t) = v_0 \mathcal{G}\left[x|s(t) - z(t), 2a^2\right], \tag{15}$$

where the symbol $\mathcal{G}[x|c,\sigma^2] = \exp\left[-(x-c)^2/(2\sigma^2)\right]$ denotes a Gaussian function of mean $c$ and variance $\sigma^2$. $u_0$, $r_0$, and $v_0$ denote the heights of the bumps of synaptic input $U(x,t)$, firing rate

$r(x,t)$, and adaptation current $V(x,t)$, respectively. $s(t)$ denotes the position (center) of the bump $U(x,t)$, which is the feature value represented by the network at time $t$. $z(t)$ denotes the delay of the adaptation current $V(t)$ to the bump $U(x,t)$, referred to as the adaptation delay hereafter.

Previous studies [47] have shown that the dynamics of a CANN is dominated by very few motion modes, and we can project the CANN dynamics on these dominating motion modes to simplify the network dynamics significantly (projecting a function $f(x,t)$ on a motion mode $u(t)$, it means to compute $\int_x f(x,t)u(x)dx$) [52]. Here, we consider the first two dominating motion modes, representing the height and position variations of the bump, respectively, which are given by,

$$\phi_0(x|s) = \mathcal{G}\left[x|s, 2a^2\right], \quad \phi_1(x|s) = (x-s)\mathcal{G}\left[x|s, 2a^2\right]. \tag{16}$$

Substituting the network state Eq.(15) into the network dynamics Eqs.(11-14), and then projecting them on the two dominating modes Eq.(16), we obtain the dynamics of the bump position $s(t)$ and the adaptation delay $z(t)$, which are written as (see SI.2 for the details),

$$\tau_s \frac{\mathrm{d}s}{\mathrm{d}t} = \frac{\gamma\Lambda}{u_0}(s^o - s) + mz, \tag{17}$$

$$\tau_z \frac{\mathrm{d}z}{\mathrm{d}t} = -z + \tau_z \frac{\mathrm{d}s}{\mathrm{d}t} + \sigma_z\sqrt{\tau_z}\xi, \tag{18}$$

where $\sigma_z = 2\sqrt{2a/(3\sqrt{3\pi})}\sigma_V/(m\sqrt{u_0})$, and $u_0 = J_0(1 + \sqrt{1 - 8\sqrt{2\pi}ak/(J_0^2\rho)})/(4\sqrt{\pi}ak)$. Eq.(18) indicates that the adaptation delay $z(t)$ in effect integrates the history of the bump position speed $(\mathrm{d}s/\mathrm{d}t)$, which enables the bump to experience history-dependent, large-step movements in the attractor space. We re-organize Eqs.(17,18) by introducing a new momentum variable $y = \Lambda(s^o - s) + (u_0/\gamma)mz$, and obtain

$$\tau_s \frac{\mathrm{d}s}{\mathrm{d}t} = \frac{\gamma}{u_0}y, \tag{19}$$

$$\tau_z \frac{\mathrm{d}y}{\mathrm{d}t} = -\frac{\tau_z}{\tau_s}\left[\left(\frac{\tau_s}{\tau_z} - m\right)\frac{u_0}{\gamma} + \Lambda\right]\frac{\gamma}{u_0}y + \Lambda(s^o - s) + \sigma_y\sqrt{\tau_z}\xi, \tag{20}$$

where $\sigma_y = 2\sqrt{2a/(3\sqrt{3\pi})}u_0\sigma_V/\gamma$. Note that the term $\Lambda(s^o - s)$ on the right-hand side of Eq.(20) comes from the external input, which conveys the stimulus information.

Comparing Eqs.(19-20) with (9-10), we see that the two dynamical systems are exactly the same when the parameters $\alpha = u_0/\gamma$ and $\beta = \tau_z/\tau_s\left[(\tau_s/\tau_z - m)u_0/\gamma + \Lambda\right]$. Moreover, by setting $\sigma_V^2 = 3\sqrt{3\pi}\gamma/(4a)(\tau_s/\tau_z - m + \gamma/u_0\Lambda)$, the condition $\sigma_y^2 = 2\beta\tau_s/\tau_z$ holds (note this condition needs not to be satisfied exactly, violating this condition for a certain amount does not affect the sampling performance, see SI.3). From Eq.(20), we also observe that for the network performing stochastic sampling, it requires $\beta < 0$, which is equivalent to the condition $m < m_{th}$, with $m_{th} \equiv \tau_s/\tau_z + \gamma\Lambda/u_0$ the threshold (if the adaptation strength $m > m_{th}$, the network bump falls into the state of moving spontaneously and no longer performs stochastic sampling; see the analysis in SI.3). It can also be checked that in the limit of $m \to 0$, the network dynamics Eqs.(17-18) returns to FLD (see the proof in SI.3). Thus, when $0 < m < m_{th}$, the network implements HDF, where the adaptation in effect plays the role of momentum.

We can also intuitively understand how the network realizes HDF from the dynamical system point of view. From Eq.(17), we see that the speed of the bump position is determined by the external input and the adaptation effect, which leads to that the momentum is given by $y = \Lambda(s^o - s) + (u_0/\gamma)mz$. In Eq.(19), the bump height $u_0$ and the strength of the external input $\gamma$ determine the difficulty of the bump movement, which leads to that the inertia (mass) of the Hamiltonian dynamics is given by $\alpha = u_0/\gamma$ (i.e., the higher the bump height $u_0$ or the smaller the input strength $\gamma$, the larger the inertia, which agree with the dynamical properties of the CANN). The friction strength of the momentum in HDF is given by $\beta = \tau_z/\tau_s\left[(\tau_s/\tau_z - m)u_0/\gamma + \Lambda\right]$, which is affected by the external input (via the term $\Lambda$, the external input tries to pin the bump position at $s^o$ which dampens the bump movement) and the adaptation strength. Specifically, $\beta$ decreases with the adaptation strength $m$ for $m < m_{th}$. This is also understandable, since adaptation increases the mobility of the bump.

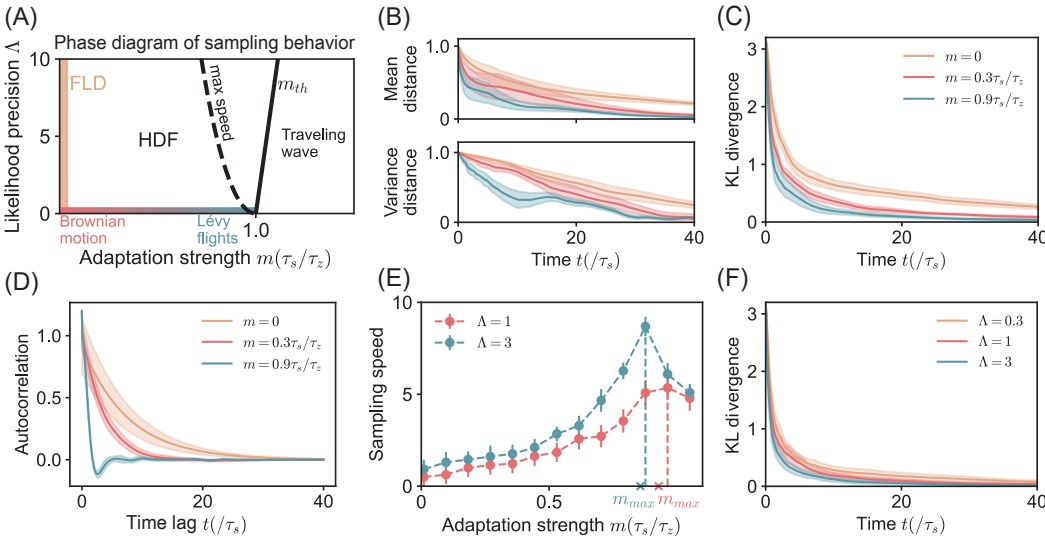

Figure 2: Noisy adaptation accelerating sampling-based inference in the CANN. (A) Sampling behaviors of the network in different parameter regimes. (B) The convergence processes of the mean (upper panel) and the variance (lower panel) of the sampled distribution to those of the target posterior, whose speeds increase with the adaptation strength for $m < m_{max}$. $\Lambda = 1$. (C) The convergence process of the KL-divergence between the sampled distribution and the target posterior, whose speed increases with the adaptation strength for $m < m_{max}$. $\Lambda = 1$. (D) The autocorrelation of sampling as a function of the time lag, whose decay speed increases with the adaptation strength for $m < m_{max}$. $\Lambda = 1$. (E) The sampling speed varies with the adaptation strength, which has the maximum value around the theoretic prediction $m_{max}$. The sampling speed is measured by the inverse of the time consuming for the KL-divergence between the sampled and the target posterior distributions reaching a small threshold $\varepsilon = 0.02$. (F) The convergence processes of the KL-divergence for different $\Lambda$. $m = 0.9\tau_s/\tau_z$. See SI.1 for parameter setting and simulation details.

## 4.3 Adaptation accelerating sampling-based inference

We further explore how exactly noisy adaptation speeds up the sampling process of the CANN. We re-organize Eqs.(17-18) as,

$$\frac{d}{dt}\begin{pmatrix} s \\ z \end{pmatrix} = -\begin{pmatrix} \gamma\Lambda/(\tau_s u_0) & -m/\tau_s \\ \gamma\Lambda/(\tau_s u_0) & (\tau_s/\tau_z - m)/\tau_s \end{pmatrix}\begin{pmatrix} s \\ z \end{pmatrix} + \begin{pmatrix} \gamma\Lambda s^o/(\tau_s u_0) \\ \gamma\Lambda s^o/(\tau_s u_0) \end{pmatrix} + \begin{pmatrix} 0 \\ \sigma_z/\sqrt{\tau_z}\xi \end{pmatrix},$$
(21)

and denote $\mathbf{H} = [\gamma\Lambda/(\tau_s u_0), -m/\tau_s; \ \gamma\Lambda/(\tau_s u_0), (\tau_s/\tau_z - m)/\tau_s]$ to be the drift matrix. It has been proved that the upper-bound of the KL-divergence between the sampled distribution $p_t(s)$ and the stationary distribution $\tilde{p}(s)$ of Eq.(21) decreases exponentially [53], i.e.,

$$KL\left[p_t(s)||\tilde{p}(s)\right] \le KL\left[p_t(s,z)||\tilde{p}(s,z)\right] \le KL\left[p_0(s,z)||\tilde{p}(s,z)\right]\exp(-ht),$$
(22)

where $p_0(s, z)$ denotes the initial distribution at $t = 0$, and $h$ denotes the smallest real-part of all eigenvalues of the drift matrix $\mathbf{H}$. Therefore, we can measure the sampling speed of HDF by the value of $h$, which is calculated to be (see SI.3)

$$h = \frac{1}{2}Re\left((m_{th} - m)/\tau_s - \sqrt{(m_{th} - m)^2/\tau_s^2 - 4\gamma\Lambda/(u_0\tau_s\tau_z)}\right),$$
(23)

where $Re(F)$ denotes the real-part of the quantity $F$. It can be checked that when $m < m_{max} \equiv (\sqrt{\tau_s/\tau_z} - \sqrt{\Lambda\gamma/u_0})^2 = m_{th} - 2\sqrt{\gamma\Lambda\tau_s/(\tau_z u_0)}$, $h$ increases with the adaptation strength $m$; when $m = m_{max}$, $h$ reaches the maximum $h_{max} = \sqrt{\gamma\Lambda/(\tau_z\tau_s u_0)}$ and the sampling reaches the fastest.

We summarize the sampling behaviours of the network in different parameter regimes (see Fig.2A):

- When no adaptation exists ($m = 0$), the CANN performs Langevin sampling.

- When no external input exists (i.e., $\Lambda = 0$, the likelihood function is uniform and contains no stimulus information), the network performs either Brownian motion (when $m$ is sufficiently small) or Lévy flights (when $m$ is close to the travelling wave boundary $\tau_s/\tau_z$) [51].

- When both external input and adaptation exist, and the adaptation strength $0 < m < m_{th}$, the network implements HDF. The sampling speed reaches the maximum when $m = m_{max}$ (the dashed line in 2A). Notably, the friction has an appropriate amplitude $\beta = 2\sqrt{\tau_z \Lambda u_0/(\gamma \tau_s)}$ when $m = m_{max}$. For $m < m_{max}$, the friction is too large which dampens the bump motion; for $m_{max} < m < m_{th}$, the friction is too small which incurs large fluctuations.

- When the adaptation strength $m > m_{th}$, the network bump falls into the state of moving spontaneously and no longer performs stochastic sampling.

## 4.4 Simulation results

We carry out simulations to validate the above theoretical analyses. In the simulations, for different values of $m$ and $\Lambda$, we fix the external feedforward input $I^{ext}(x)$ given by Eq.14 (which conveys the likelihood function of the observation) and evolve the network dynamics Eqs.(11-13) to sample the stimulus feature value. The results are presented in Figs.1-2.

Firstly, we observe that the network dynamics indeed achieves sampling-based inference. As shown in Fig.1E, because of noises, the bump position of the network fluctuates with time, whose trace samples the stimulus feature value in the attractor space. Over time, the stationary sampled distribution $\tilde{p}(s)$ approaches the target posterior distribution $p(s|s^\circ)$ as shown in Fig.1F.

Secondly, we observe that noisy adaptation indeed accelerates the sampling process. As shown in Fig.2B, the mean and the variance of the sampled distribution approach to those of the target posterior distribution asymptotically as time goes on, and the converging process is speed up when the adaptation strength $m$ increases, for $0 \leq m < m_{max}$. This is further confirmed by the converging process of the KL-divergence between the sampled distribution and the target posterior as shown in Fig.2C. Fig.2D presents autocorrelation as a different measure to demonstrate that the adaptation speeds up the sampling, for $0 \leq m < m_{max}$. Fig.2E displays how the sampling speed varies with the adaptation strength, which reaches the maximum around the theoretically predicted point $m_{max}$. Fig.2F confirms that the network implements rapid sampling for different input uncertainties.

## 5 Coupled CANNs with noisy adaptation accelerating distributed sampling-based inference

We extend the above study to coupled CANNs with noisy adaptation (Fig.3A). The coupled CANNs have been used to model multi-sensory integration between brain regions. Previous studies have revealed that reciprocally connected CANNs, with each of them modelling the representation of heading direction in one brain region, can achieve distributed Bayesian inference [38, 54]. Here, we derive that coupled CANNs with noisy adaptation implement HDF, which accelerates sampling-based inference in the high-dimensional feature space. The dynamics of the $i$th network in coupled CANNs is given by,

$$\tau_s \frac{\partial U_i(x,t)}{\partial t} = -U_i(x,t) + \rho \int_{x'} W_i(x,x')r_i(x',t)\mathrm{d}x' + \rho \sum_{j\neq i}^{M} \int_{x'} \tilde{W}_{ij}(x,x')r_j(x',t)\mathrm{d}x'$$
$$+ \gamma I_i^{ext}(x) - V_i(x,t), \tag{24}$$

where $W_i(x,x') = J_i/(\sqrt{2\pi}a) \exp\left[-(x-x')^2/(2a^2)\right]$ denote the recurrent connections between neurons in CANN $i$, and $\tilde{W}_{ij}(x,x') = G_{ij}/(\sqrt{2\pi}a) \exp\left[-(x-x')^2/(2a^2)\right]$, for $j \neq i$, denote the reciprocal connections from neurons in CANN $j$ to neurons in CANN $i$. The $i$th CANN receives the external input $I_i^{ext}(x) = \Lambda_i \exp\left[-(x-s_i^\circ)^2/(4a^2)\right]$, which conveys the information of the feature $s_i$, with $s_i^\circ$ the corresponding observation and $\Lambda_i$ the precision. The dynamics of the neuronal firing rate and the adaptation current in each CANN have the same forms as in Eq.(12) and (13), respectively. Adaptation noises between different CANNs are independent to each other, i.e., $\langle \xi_i(x,t)\xi_j(x',t')\rangle = \delta_{ij}\delta(t-t')\delta(x-x')$.

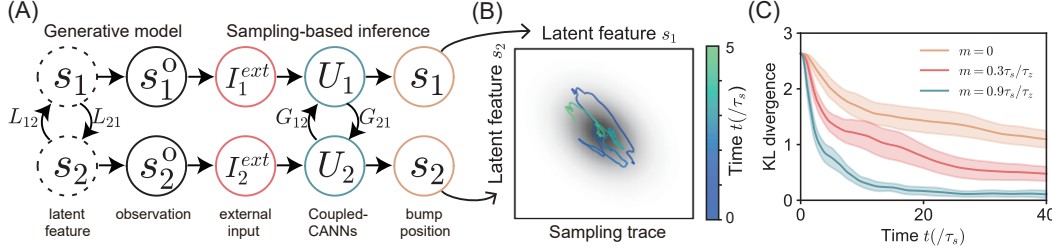

Figure 3: Coupled CANNs with noisy adaptation implement distributed sampling-based Bayesian inference. (A) The sampling-based inference in two coupled CANNs. (B) Example sampling traces in the two-dimensional feature space. (C) The convergence process of the KL-divergence between the sampled distribution $p(\mathbf{s})$ and the target posterior $p(\mathbf{s}|\mathbf{s}^\mathrm{o})$, whose speed increases with the adaptation strength $m$, for $0 < m < m_{max}$. See SI.1 for parameter setting and simulation details.

Again, by assuming that the state of each CANN has the Gaussian forms as in Eq.(15) and projecting the network dynamics onto the two dominating modes Eq.(16), we obtain the dynamics of bump positions $\mathbf{s}(t)$ and the corresponding adaptation delays $\mathbf{z}(t)$, which are written as (see details in SI.4),

$$\tau_s \frac{d\mathbf{s}}{dt} = \gamma \mathbf{u}^{-1} \left[ \mathbf{\Lambda} \mathbf{s}^\mathrm{o} - \left( \frac{\mathbf{u}}{\gamma} \mathbf{J}^{-1} \mathbf{G} + \mathbf{\Lambda} \right) \mathbf{s} \right] + m\mathbf{z}, \tag{25}$$

$$\tau_z \frac{d\mathbf{z}}{dt} = -\mathbf{z} + \tau_z \frac{d\mathbf{s}}{dt} + \sqrt{\tau_z} \boldsymbol{\sigma}_z \boldsymbol{\xi}, \tag{26}$$

where $\mathbf{s} = \{s_i\}$, $\mathbf{z} = \{z_i\}$, $\mathbf{J} = \mathrm{diag}\{J_i\}$ and $\mathbf{\Lambda} = \mathrm{diag}\{\Lambda_i\}$, for $i = 1, \dots, M$, and $\mathbf{G} = \{G_{ij}\}$ is a Laplacian matrix. $\mathbf{u} = \mathrm{diag}\{u_i\}$, with $u_i = J_i(1 + \sqrt{1 - 8\sqrt{2\pi}ak/(J_i^2 \rho)})/(4\sqrt{\pi}ak)$ representing the bump height of CANN $i$. $\boldsymbol{\xi}$ denote Gaussian white noises satisfying $\langle \boldsymbol{\xi}(t)\boldsymbol{\xi}^T(t') \rangle = \mathbf{I}\delta(t - t')$, and the noise strengths $\boldsymbol{\sigma}_z \boldsymbol{\sigma}_z^T = 8a/(3\sqrt{3\pi}m)\sigma_V^2 \mathbf{u}^{-1}$.

By introducing a new set of momentum variables $\boldsymbol{y} = \mathbf{\Lambda}\mathbf{s}^\mathrm{o} - \left( \mathbf{u}\mathbf{J}^{-1}\mathbf{G}/\gamma + \mathbf{\Lambda} \right)\mathbf{s} + m\mathbf{u}\mathbf{z}/\gamma$, Eqs.(25-26) are re-organized as,

$$\tau_s \frac{d\mathbf{s}}{dt} = \gamma \boldsymbol{u}^{-1} \boldsymbol{y} \tag{27}$$

$$\tau_z \frac{d\mathbf{y}}{dt} = -\frac{\tau_z}{\tau_s} \left[ \left( \frac{\tau_s}{\tau_z} - m \right) \frac{\mathbf{u}}{\gamma} + \frac{\mathbf{u}}{\gamma}\mathbf{J}^{-1}\mathbf{G} + \mathbf{\Lambda} \right] \gamma \boldsymbol{u}^{-1}\boldsymbol{y} + \mathbf{\Lambda}\mathbf{s}^\mathrm{o} - (\frac{\mathbf{u}}{\gamma}\mathbf{J}^{-1}\mathbf{G} + \mathbf{\Lambda})\mathbf{s}$$
$$+ \boldsymbol{\sigma}_y \sqrt{\tau_z}\boldsymbol{\xi}, \tag{28}$$

where $\boldsymbol{\sigma}_y = 2\sqrt{2a\mathbf{u}/(3\sqrt{3\pi})}\sigma_V/\gamma$. Compared to Eqs.(7-8), we see that by setting $\boldsymbol{\alpha} = \mathbf{u}/\gamma$, $\boldsymbol{\beta} = \tau_z/\tau_s \left[ (\tau_s/\tau_z - m)\mathbf{u}/\gamma + \mathbf{L} + \mathbf{\Lambda} \right]$, and the reciprocal connections between CANNs,

$$\mathbf{L} = \frac{\mathbf{u}}{\gamma}\mathbf{J}^{-1}\mathbf{G}, \tag{29}$$

the coupled CANNs with noisy adaptation implement HDF, and they sample the posterior distribution given by Eqs.(4-5) (Fig.3A; see SI.4). Notably, Eq.(29) shows that the correlation prior of $\mathbf{s}$ (the Laplacian matrix $\mathbf{L}$ in Eq.(3)) is stored in the reciprocal connections between coupled CANNs, i.e., $\mathbf{G} = \gamma \mathbf{J}\mathbf{u}^{-1}\mathbf{L}$. The sampling dynamics Eqs.(25-26) achieve that the sampled marginal distribution of each feature $p(s_i)$ by each CANN equals to the corresponding marginal target posterior $p(s_i|\mathbf{s}^\mathrm{o})$, implying that coupled CANNs realize Bayesian inference in a distributed way (see SI.4).

Following the same calculation as in the one dimensional case, we quantify the sampling speed of coupled CANNs by computing the smallest real-part of all eigenvalues of the drift matrix in Eqs.(27-28), which is given by (see SI.5)

$$h = \frac{1}{2}Re \left( 1/\tau_z - m/\tau_s + Q/\tau_s - \sqrt{\left[1/\tau_z - m/\tau_s + Q/\tau_s\right]^2 - 4Q/(\tau_s \tau_z)} \right). \tag{30}$$

where $Q$ is the smallest eigenvalue of the matrix $\boldsymbol{\alpha}^{-1}(\mathbf{L} + \mathbf{\Lambda})$. It can checked that when $m = (\sqrt{\tau_s/\tau_z} - \sqrt{Q})^2$, the coupled CANNs achieves the fastest sampling speed. We carry out simulations to confirm the above theoretical analyses, and the results are shown in Fig.3.

# 6 Conclusions and discussions

The present study explored how sampling-based Bayesian inference of continuous variables in CANNs is accelerated. We theoretically derived how noisy adaptation enables a CANN to implement HDF, and elucidate that the adaptation effectively plays the role of momentum to speed up the sampling process. We systematically analyzed the sampling performances of the network and derived the condition when the adaptation has the maximum acceleration effect. All theoretical analyses are validated by simulation results. Furthermore, we studied coupled CANNs, where the reciprocal connections between CANNs store the prior correlations between features, and we showed that noisy adaptation accelerates distributed sampling in the high-dimensional feature space.

Sampling-based inference is a promising strategy to realize Bayesian inference in the brain, being consistent with the stochastic responses of neurons [17, 18, 20]. Several studies have investigated the issue of accelerating stochastic sampling in neural circuits. However, these studies have typically considered feature representation at the individual neuron level or their linear combination, and the sampling acceleration relies on asymmetric connections between neurons [15] or redundant representation of a neuron group [25]. Here, we consider that a continuous feature is represented by a neuron ensemble jointly in the form of a CANN and the sampling acceleration relies on the adaptation of neuronal responses. Nevertheless, all these proposed mechanisms are not necessarily exclusive to each other. Rather, they are more likely complementary to each other to coherently speed up sampling at various levels, which collaboratively implement hierarchical Bayesian inference in the brain. In order to elucidate the mechanism of accelerating sampling by adaptation analytically, we have only considered a linear Gaussian generative model, while neural systems may infer latent variables from very complicated generative processes and their posterior can be multimodal. We will therefore extend the current study to nonlinear generative models, such as the Gaussian scale mixture model [17, 18], in the future work. We hope this study enhances our understanding of the sampling-based Bayesian inference in the brain.

**On the stochasticity of the adaptation dynamics.** In our model, spike frequency adaptation (SFA) is used to implement adaptation. A number of mechanisms in biological systems can realize SFA, and three of them are often studied [45], which are: 1) the current caused by voltage-dependent, high-threshold potassium channels; 2) the current mediated by calcium-dependent potassium channels; 3) the current caused by the slow recovery from in-activation of fast sodium channels. All these mechanisms depend on ion concentrations, release of neural transmitters, activation/inactivation of ion channels, buffering and diffusion, and all these processes are very noisy (see Fig.9 in [55]). Recent works also show that adaptation noises can play important computational roles (see, e.g., [56]). Indeed, previous modeling works rarely consider adaptation noises, since they focused on different things rather than the functions of adaptation noises. Here, we show that adaption noises can actually contribute to accelerate stochastic sampling.

**Time scales observed in neuroscience experiments.** In neuroscience society, it remains unknown yet the exact time cost for the brain performing Bayesian inference. The only known fact is that this process is extremely fast [57]. In monkey experiments, the electrophysiology studies revealed that the visual cortex can accomplish a probabilistic decision-making task in less than 800ms [58] and a contour integration task in less than 200ms [59]. If we plug in the biological relevant parameter value 5-10ms in the model, it gives that the sampling speed of HDF is around 100-200ms, while the speed of FLD is around 2-4s in a single CANN (see Fig.2B-C). Hence, HDF is about 20 times faster than FLD.

## Acknowledgement

This work was supported by Science and Technology Innovation 2030-Brain Science and Brain-inspired Intelligence Project (No.2021ZD0200204), Guangdong Province with Grant (No.2018B030338001), the National Natural Science Foundation of China (No.4861425025, T.J.Huang), and Beijing Academy of Artificial Intelligence.

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
