# Adaptation Accelerating Sampling-based Bayesian Inference in Attractor Neural Networks: Supplementary Information

**Xingsi Dong**[1]
dxs19980605@pku.edu.cn

**Zilong Ji**[1,3]
zilong.ji@ucl.ac.uk

**Tianhao Chu**[1]
chutianhao@stu.pku.edu.cn

**Tiejun Huang**[4]
tjhuang@pku.edu.cn

**Wen-Hao Zhang**[2,†]
wenhao.zhang@utsouthwestern.edu

**Si Wu**[1,†]
siwu@pku.edu.cn

1, School of Psychology and Cognitive Sciences, IDG/McGovern Institute for Brain Research,
PKU-Tsinghua Center for Life Sciences, Academy for Advanced Interdisciplinary Studies,
Center of Quantitative Biology, Peking University.
2. Lyda Hill Department of Bioinformatics, O'Donnell Brain Institute, UT Southwestern Medical Center.
3. Institute of Cognitive Neuroscience, University College London
4. School of Computer Science, Peking University.
[†]: Corresponding authors.

## 1 Parameter setting and simulation details

### 1.1 Parameter setting

Table 1 lists the parameters used in Fig. 1&2 in the main text; Table 2 lists the parameters used in Fig. 3 in the main text.

Table 1: Parameters used in Fig. 1&2 in the main text.

| Parameters | Value |
|---|---|
| Time constant of $U$: $\tau_s$ | 1 |
| Time constant of $V$: $\tau_z$ | 5 |
| Neuron density: $\rho$ | 1 |
| Global inhibition strength: $k$ | 0.5 |
| Recurrent connection strength: $J_0$ | 10 |
| Recurrent connection radius: $a$ | $\pi/10$ |
| Input strength: $\gamma$ | 0.1 |
| Observation: $s^{\mathrm{o}}$ | 0 |

### 1.2 Simulation details

#### 1.2.1 For Fig. 1& 2

In the simulation, the periodic boundary $(-\pi, \pi]$ is used for the feature space. The CANN contains $N = 360$ neurons uniformly distributed in the feature space. Other parameters are listed in Table.1. For fixed values of $m$ and $\Lambda$, we simulate the network dynamics for 50 trials. In a single trial, the network dynamics is simulated using the Euler method with time step $\Delta t = 0.01\tau_s$. We collect the traces of bump position to calculate the sampled mean, variance, distribution, and autocorrelation.

36th Conference on Neural Information Processing Systems (NeurIPS 2022).

Table 2: Parameters used in Fig. 3 in the main text.

| Parameters | Value |
|---|---|
| Dimension of feature $\mathbf{s}$ (number of CANNs): $M$ | 5 |
| Time constant of $U_i$: $\tau_s$ | 1 |
| Time constant of $V_i$: $\tau_z$ | 5 |
| Neuron density: $\rho$ | 1 |
| Global inhibition strength: $k$ | 0.5 |
| Recurrent connection strength: $J_i$ | 10 |
| Recurrent connection radius: $a$ | $\pi/10$ |
| Input strength: $\gamma$ | 0.1 |
| Observation: $\mathbf{s}^o$ | $\mathbf{0}$ |
| Elements of prior matrix: $\mathbf{L}$ | randomly sampled in range $[-1, 0)$ |
| Elements of likelihood matrix: $\mathbf{\Lambda}$ | randomly sampled in range $(0, 1]$ |

### 1.2.2 For Fig. 3

Each CANN in coupled-CANNs is the same as the single CANN case described above. The periodic boundary $(-\pi, \pi]$ is used for each feature $s_i$. Each CANN contains $N = 360$ neurons uniformly distributed in its feature space. Other parameters are listed in Table.2. The connection strengths between CANNs are calculated by Eq.(29) in the main text. For a fixed value of $m$, we simulate the network dynamics for 50 trials. In a single trial, the network dynamics is simulated by using the Euler method with time step $\Delta t = 0.01\tau_s$. We collect the traces of bump position to calculate the sampled distribution.

## 2  Sampling dynamics of a 1D CANN with noisy adaptation

In this section, we present the mathematical details of using a projection method to derive the dynamics of the bump position $s(t)$ and the adaptation delay $z(t)$ of the network, i.e, Eq.(17-18) in the main text.

As shown in the main text (Eq.11-14), the dynamics of a CANN with adaptation is written as,

$$\tau_s \frac{\partial U(x,t)}{\partial t} = -U(x,t) + \rho \int_{x'} W(x,x')r(x',t)\mathrm{d}x' + \gamma I^{ext}(x,t) - V(x,t), \qquad (\text{S1})$$

$$\tau_z \frac{\partial V(x,t)}{\partial t} = -V(x,t) + mU(x,t) + \sigma_V \sqrt{\tau_z U(x,t)}\xi(x,t), \qquad (\text{S2})$$

$$r(x,t) = \frac{U^2(x,t)}{1 + k\rho \int_{x'} U^2(x',t)\mathrm{d}x'}. \qquad (\text{S3})$$

where $W(x,x') = J_0 \exp\left[-(x-x')/(2a^2)\right]$ and $I^{ext}(x,t) = \gamma\Lambda \exp\left[-(x-s^o)^2/(4a^2)\right]$. And $\xi(x,t)$ are gaussian white noise satisfying $\langle \xi(x,t) \rangle = 0$ and $\langle \xi(x,t)\xi(x',t') \rangle = \delta(t-t')\delta(x-x')$.

As shown in the main text (Eq.15), the presumed network state have the following form,

$$U(x,t) = u_0 \exp\left[-\frac{(x-s)^2}{4a^2}\right], \qquad (\text{S4})$$

$$r(x,t) = r_0 \exp\left[-\frac{(x-s)^2}{2a^2}\right], \qquad (\text{S5})$$

$$V(x,t) = v_0 \exp\left[-\frac{(x-s+z)^2}{4a^2}\right]. \qquad (\text{S6})$$

The first two dominating motion modes representing the height and position variations of the bump are given by (Eq.16 in the main text),

$$\phi_0(x|s) = \exp\left[-\frac{(x-s)^2}{4a^2}\right],$$ (S7)

$$\phi_1(x|s) = (x-s)\exp\left[-\frac{(x-s)^2}{4a^2}\right].$$ (S8)

Substituting Eqs.(S4-S5) into (S3), we get the relationship between $r_0$ and $u_0$, which is,

$$r_0 = \frac{u_0^2}{1 + k\rho\sqrt{2\pi}au_0^2}.$$ (S9)

Substituting Eqs.(S4-S6) into (S1), we get,

$$\tau_s u_0 \frac{\mathrm{d}}{\mathrm{d}t}\exp\left[-\frac{(x-s)^2}{4a^2}\right] = \left(-u_0 + \frac{\rho J_0}{\sqrt{2}}r_0\right)\exp\left[-\frac{(x-s)^2}{4a^2}\right] + \gamma\Lambda\exp\left[-\frac{(x-s^{\mathrm{o}})^2}{4a^2}\right]$$
$$- v_0\exp\left[-\frac{(x-s+z)^2}{4a^2}\right].$$ (S10)

Projecting both sides of the above equation onto the motion mode $\phi_0(x|s)$, we obtain,

$$0 = -u_0 + \frac{\rho J_0}{\sqrt{2}}r_0 + \gamma\Lambda\exp\left[-\frac{(s^{\mathrm{o}}-s)^2}{8a^2}\right] - v_0\exp\left(-\frac{z^2}{8a^2}\right).$$ (S11)

Here, projecting a function $f(x,t)$ on a motion mode $u(t)$ means to compute $\int_x f(x,t)u(x)\mathrm{d}x$.

Projecting both sides onto the motion mode $\phi_1(x|s)$, we obtain,

$$\tau_s u_0 \frac{\mathrm{d}s}{\mathrm{d}t} = \gamma\Lambda(s^{\mathrm{o}}-s)\exp\left[-\frac{(s^{\mathrm{o}}-s)^2}{8a^2}\right] + v_0 z\exp\left(-\frac{z^2}{8a^2}\right).$$ (S12)

Substituting Eqs.(S4-S6) into (S2), we get,

$$\tau_z v_0 \frac{\mathrm{d}}{\mathrm{d}t}\exp\left[-\frac{(x-s+z)^2}{4a^2}\right] = -v_0\exp\left[-\frac{(x-s+z)^2}{4a^2}\right] + mu_0\exp\left[-\frac{(x-s)^2}{4a^2}\right]$$
$$+ \sigma_V\sqrt{\tau_z u_0}\exp\left[-\frac{(x-s)^2}{8a^2}\right]\xi(x,t).$$ (S13)

Projecting both sides of the above equation onto the motion mode $\phi_0(x|s)$, we obtain,

$$\tau_z \frac{z}{4a^2}v_0\exp\left(-\frac{z^2}{8a^2}\right)\frac{\mathrm{d}s}{\mathrm{d}t} = -v_0\exp\left(-\frac{z^2}{8a^2}\right) + mu_0 + \sqrt{\frac{1}{a\sqrt{3\pi}}}\sigma_V\sqrt{\tau_z u_0}\xi_0.$$ (S14)

where $\xi_0$ is Gaussian white noise of zero mean and unit variance.

Projecting both sides onto the motion mode $\phi_1(x|s)$, we obtain,

$$\tau_z v_0\exp\left(-\frac{z^2}{8a^2}\right)\left(\frac{1}{2}-\frac{z^2}{8a^2}\right)\left(\frac{\mathrm{d}s}{\mathrm{d}t}-\frac{\mathrm{d}z}{\mathrm{d}t}\right) = \frac{v_0 z}{2}\exp\left(-\frac{z^2}{8a^2}\right) + \sqrt{\frac{2a}{3\sqrt{3\pi}}}\sigma_V\sqrt{\tau_z u_0}\xi_1.$$ (S15)

where $\xi_1$ is Gaussian white noise of zero mean and unit variance.

Utilizing the properties $z^2 \ll 8a^2$ and $\gamma \ll J_0$, and solving Eqs.(S9,S11,S14), we obtain,

$$u_0 = \frac{J_0}{4\sqrt{\pi}ak}\left(1 + \sqrt{1 - \frac{8\sqrt{2\pi}ak}{J_0^2\rho}}\right),$$ (S16)

$$v_0 = mu_0.$$ (S17)

Further solving Eqs.(S12,S15), we obtain,

$$\tau_s\frac{\mathrm{d}s}{\mathrm{d}t} = \frac{\gamma\Lambda}{u_0}(s^{\mathrm{o}}-s) + mz,$$ (S18)

$$\tau_z\frac{\mathrm{d}z}{\mathrm{d}t} = -z + \tau_z\frac{\mathrm{d}s}{\mathrm{d}t} + \sqrt{\tau_z}\sigma_z\xi_1.$$ (S19)

where $\sigma_z = 2\sqrt{2a/(3\sqrt{3\pi})}\sigma_V/(m\sqrt{u_0})$. The above dynamics gives Eq.(17-18) in the main text.

# 3 Sampling performance of a 1D CANN with noisy adaptation

In this section, we present the detailed analyses of the sampling performance of a 1D CANN with noisy adaptation.

## 3.1 Sampling performance of the network

We re-organize Eqs.(S18-S19) to be,

$$\frac{\mathrm{d}}{\mathrm{d}t}\begin{pmatrix} s \\ z \end{pmatrix} = -\begin{pmatrix} \gamma\Lambda/(\tau_s u_0) & -m/\tau_s \\ \gamma\Lambda/(\tau_s u_0) & (\tau_s/\tau_z - m)/\tau_s \end{pmatrix}\begin{pmatrix} s \\ z \end{pmatrix} + \begin{pmatrix} \gamma\Lambda s^{\mathrm{o}}/(\tau_s u_0) \\ \gamma\Lambda s^{\mathrm{o}}/(\tau_s u_0) \end{pmatrix} + \begin{pmatrix} 0 \\ \sigma_z/\sqrt{\tau_z}\xi \end{pmatrix},$$

(S20)

and denote $\mathbf{H} = [\gamma\Lambda/(\tau_s u_0), -m/\tau_s; \; \gamma\Lambda/(\tau_s u_0), (\tau_s/\tau_z - m)/\tau_s]$ to be the drift matrix.

The eigenvalues ($\lambda$) of the drift matrix determine the behavior of the dynamic system, which are calculated by,

$$\begin{vmatrix} \gamma\Lambda/(\tau_s u_0) - \lambda & -m\tau_s^{-1} \\ \gamma\Lambda/(\tau_s u_0) & \tau_z^{-1} - m\tau_s^{-1} - \lambda \end{vmatrix} = 0,$$

(S21)

which gives

$$\lambda^{\pm} = \frac{1}{2}\left((\tau_s/\tau_z + \Lambda\gamma/u_0 - m)/\tau_s \pm \sqrt{(\tau_s/\tau_z + \Lambda\gamma/u_0 - m)^2/\tau_s^2 - 4\gamma\Lambda/(u_0\tau_s\tau_z)}\right).$$

(S22)

The real part of the smallest eigenvalue $h$ determines the convergence of the dynamic system, which is calculated as $h = \min\left(\mathrm{Re}(\lambda^-), \mathrm{Re}(\lambda^+)\right) = \mathrm{Re}(\lambda^-)$, and it gives Eq.(23) in the main text. It is straightforward to check that:

- When $0 < m \leq m_{max} = (\sqrt{\tau_s/\tau_z} - \sqrt{\Lambda\gamma/u_0})^2$, $h$ monotonically increases with $m$ and $h > 0$.
- When $m > m_{max}$, $h$ monotonically decreases with $m$. In particular, when $m_{max} < m < m_{th} = \tau_s/\tau_z + \Lambda\gamma/u_0$, $h > 0$; when $m > m_{th}$, $h < 0$, indicating the divergence of the dynamic system.

Thus, the network performs HDF when $0 < m < m_{th}$, and when $m = m_{max}$, $h$ reaches the maximum value, i.e., the sampling reaches the fastest speed.

## 3.2 Returning to FLD when $m \to 0$

We show that Eq.(S18-S19) degenerates to FLD when $m$ is sufficently small. When $m \to 0$, Eq.(S18) shows that the variation of $s$ is rather slow when it approaches to the stationary distribution. And because of $\sigma_z = 2\sqrt{2a/(3\sqrt{3\pi})}\sigma_V/(m\sqrt{u_0}) \to \infty$, the delay variable $z$ changes much faster than $s$ which can be regarded as a fast variable. Therefore, we can regard $s$ as fixed and approximate Eq.(S19) to be,

$$\tau_z\frac{\mathrm{d}z}{\mathrm{d}t} = -z + \sqrt{\tau_z}\sigma_z\xi$$

(S23)

which gives that the stationary distribution of $z$ to be Gaussian, i.e., $\tilde{p}(z) = \mathcal{N}\left(0, \sigma_z\sigma_z^T/(2\tau_z)\right)$. Thus, by setting $\mathrm{d}t = 2\tau_z^2$, Eq.(S18) can be written as,

$$\tau_s\frac{\mathrm{d}s}{\mathrm{d}t} = \frac{\gamma}{u_0}\Lambda(s^{\mathrm{o}} - s) + \sqrt{\tau_s}\sigma_s\xi_s,$$

(S24)

where $\sigma_s = m\sigma_z\sqrt{\mathrm{d}t/(2\tau_z\tau_s)}$, which implements FLD, i.e., Eq.(6) in the main text.

## 3.3 The effect of $\sigma_V^2$

As stated in the main text, it in theory requires the condition $\sigma_V^2 = \sigma_{opt}^2 \equiv 3\sqrt{3\pi}\gamma(\tau_s/\tau_z - m + \Lambda\gamma/u_0)/(4a)$, for the stationary distribution $\tilde{p}(s)$ of Eq.(S18-S19) equalling to the target distribution $p(s|s^{\mathrm{o}})$. We check in practice how restricted this condition is for the network to have a good performance.

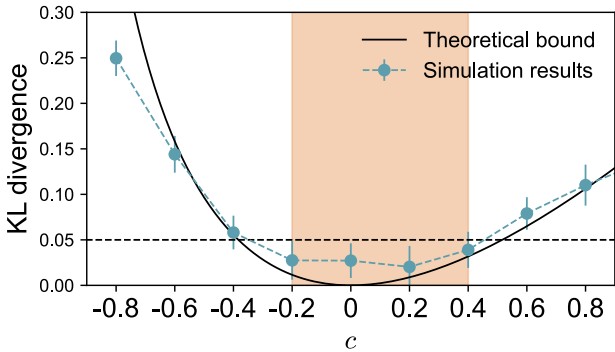

Figure S1: The KL divergence between the stationary and target distributions vs. the violation of the optimal noise strength.

We set $\sigma_V^2 = (1+c)\sigma_{opt}^2$, with $c$ controlling the violation of the optimal condition. It can be proved that the KL divergence between the stationary and the target distribution is bounded by

$$\mathrm{KL}\left[\tilde{p}(s)||p(s|s^o)\right] \leq \frac{1}{2}\left[c - \ln(1+c)\right].\tag{S25}$$

As shown in Fig. S1, simulation results agree well with the theoretical bound (Eq. S25). The performance of the network is robust for a wide range of $\sigma_V^2$ values (orange area in Fig. S1): up to $-20\%$ or $40\%$ violation of the optimal noise strength, the KL-divergence between the stationary and the target posterior distributions is smaller than $0.05$. Furthermore, considering that $\gamma \ll u_0$, $\sigma_V^2 \approx 3\sqrt{3\pi}\gamma(\tau_s/\tau_z - m)/(4a)$, which is independent of the input uncertainty $\Lambda$.

## 4   Sampling dynamics of coupled CANNs

In this section, we present the mathematical details of using a projection method to derive the sampling dynamics of coupled CANNs, i.e., Eqs.(25-16) in the main text.

As described in the main text (Eq.24 and the followed descriptions), the dynamic of coupled CANNs with noisy adaptation are written as,

$$\tau_s\frac{\partial U_i(x,t)}{\partial t} = -U_i(x,t) + \rho\int_{x'}W_i(x,x')r_j(x',t)\mathrm{d}x' + \rho\sum_{j\neq i}^{M}\int_{x'}\tilde{W}_{ij}(x,x')r_j(x',t)\mathrm{d}x'$$
$$+ \gamma I_i^{ext}(x,t) - V_i(x,t),\tag{S26}$$

$$\tau_z\frac{\partial V_i(x,t)}{\partial t} = -V_i(x,t) + mU_i(x,t) + \sigma_V\sqrt{\tau_z U_i(x,t)}\xi_i(x,t),\tag{S27}$$

$$r_i(x,t) = \frac{U_i^2(x,t)}{1 + k\rho\int_{x'}U_i^2(x',t)\mathrm{d}x'},\tag{S28}$$

where $W_i(x,x') = J_i\exp\left[-(x-x')/(2a^2)\right]$, $\tilde{W}_{ij}(x,x') = G_{ij}\exp\left[-(x-x')/(2a^2)\right]$, $I_i^{ext}(x,t) = \gamma\Lambda_i\exp\left[-(x-s_i^o)^2/(4a^2)\right]$, $\langle\xi_i(x,t)\rangle = 0$ and $\langle\xi_i(x,t)\xi_j(x',t')\rangle = \delta_{ij}\delta(t-t')\delta(x-x')$.

The state of each CANN is assumed to have the following form,

$$U_i(x,t) = u_i\exp\left[-\frac{(x-s_i)^2}{4a^2}\right],\tag{S29}$$

$$r_i(x,t) = R_i\exp\left[-\frac{(x-s_i)^2}{2a^2}\right],\tag{S30}$$

$$V_i(x,t) = v_i\exp\left[-\frac{(x-s_i+z_i)^2}{4a^2}\right].\tag{S31}$$

The first two dominating motion modes representing the height and position variations of the bump are,

$$\phi_0(x|s_i) = \exp\left[-\frac{(x-s_i)^2}{4a^2}\right],$$ (S32)

$$\phi_1(x|s_i) = (x-s_i)\exp\left[-\frac{(x-s_i)^2}{4a^2}\right].$$ (S33)

Substituting Eqs.(S29-S30) into (S28), we can get the relationship between $R_i$ and $u_i$, which is

$$R_i = \frac{u_i^2}{1 + k\rho\sqrt{2\pi}au_i^2}.$$ (S34)

Substituting Eqs.(S29-S31) into (S26), we get

$$\tau_s u_i \frac{d}{dt}\exp\left[-\frac{(x-s_i)^2}{4a^2}\right] = -u_i\exp\left[-\frac{(x-s_i)^2}{4a^2}\right] + \frac{\rho}{\sqrt{2}}J_i R_i\exp\left[-\frac{(x-s_i)^2}{4a^2}\right]$$
$$+ \frac{\rho}{\sqrt{2}}\sum_{j\neq i}^{M} G_{ij}R_j\exp\left[-\frac{(x-s_j)^2}{4a^2}\right] + \gamma\Lambda_i\exp\left[-\frac{(x-s_i^o)^2}{4a^2}\right] - v_i\exp\left[-\frac{(x-s_i+z_i)^2}{4a^2}\right].$$ (S35)

Projecting both sides onto the motion mode $\phi_0(x|s_i)$, we obtain

$$0 = -u_i + \frac{\rho}{\sqrt{2}}J_i R_i + \frac{\rho}{\sqrt{2}}\sum_{j\neq i}^{M}G_{ij}R_j\exp\left[-\frac{(s_i-s_j)^2}{8a^2}\right] + \gamma\Lambda_i\exp\left[-\frac{(s_i^o-s_i)^2}{8a^2}\right]$$
$$- v_i\exp\left(-\frac{z_i^2}{8a^2}\right)$$ (S36)

Projecting both sides onto the motion mode $\phi_1(x|s_i)$, we obtain

$$\tau_s u_0\frac{ds_i}{dt} = \frac{\rho}{\sqrt{2}}\sum_{j\neq i}^{M}G_{ij}R_j(s_j-s_i)\exp\left[-\frac{(s_i-s_j)^2}{8a^2}\right] + \gamma\Lambda_i(s_i^o-s_i)\exp\left[-\frac{(s_i^o-s_i)^2}{8a^2}\right]$$
$$+ v_i z_i\exp\left(-\frac{z_i^2}{8a^2}\right).$$ (S37)

Substituting Eqs.(S29-S31) into (S27), we get

$$\tau_z v_i\frac{d}{dt}\exp\left[-\frac{(x-s_i+z_i)^2}{4a^2}\right] = -v_i\exp\left[-\frac{(x-s_i+z_i)^2}{4a^2}\right] + mu_i\exp\left[-\frac{(x-s_i)^2}{4a^2}\right]$$
$$+ \sigma_V\sqrt{\tau_z u_i}\exp\left[-\frac{(x-s_i)^2}{8a^2}\right]\xi_i(x,t).$$ (S38)

Projecting both sides onto the motion mode $\phi_0(x|s_i)$, we obtain

$$\tau_z\frac{z_i}{4a^2}v_i\exp\left(-\frac{z_i^2}{8a^2}\right)\frac{ds_i}{dt} = -v_i\exp\left(-\frac{z_i^2}{8a^2}\right) + mu_i + \sqrt{\frac{1}{a\sqrt{3\pi}}}\sigma_V\sqrt{\tau_z u_i}\xi_{i,0}.$$ (S39)

Projecting both sides onto the motion mode $\phi_1(x|s)$, we obtain

$$\tau_s v_i\exp\left(-\frac{z_i^2}{8a^2}\right)\left(\frac{1}{2}-\frac{z_i^2}{8a^2}\right)\left(\frac{ds_i}{dt}-\frac{dz_i}{dt}\right) = \frac{v_i z_i}{2}\exp\left(-\frac{z_i^2}{8a^2}\right) + \sqrt{\frac{2a}{3\sqrt{3\pi}}}\sigma_V\sqrt{\tau_z u_i}\xi_{i,1}.$$ (S40)

The noise terms $\xi_{i,0}$ and $\xi_{i,1}$ are written as

$$\xi_{i,0}(t) = \frac{1}{\sqrt{\sqrt{2\pi}a}}\int\exp\left[-\frac{(x-s_i)^2}{4a^2}\right]\xi_i(x,t)dx,$$ (S41)

$$\xi_{i,1}(t) = \frac{1}{\sqrt{\sqrt{2\pi}a^3}}\int(x-s_i)\exp\left[-\frac{(x-s_i)^2}{4a^2}\right]\xi_i(x,t)dx.$$ (S42)

It can be checked that

$$\langle \xi_{i,1}(t) \rangle = 0, \tag{S43}$$

$$\langle \xi_{i,1}(t)\xi_{j,1}(t') \rangle = \delta_{ij}\delta(t - t'). \tag{S44}$$

Utilizing the properties $z_i^2 \ll 8a^2$ and $\gamma \ll J_i$, and solving Eqs.(S34,S36,S39), we obtain

$$u_i = \frac{J_i}{4\sqrt{\pi}ak}\left(1 + \sqrt{1 - \frac{8\sqrt{2\pi}ak}{J_i^2\rho}}\right), \tag{S45}$$

$$v_i = mu_i. \tag{S46}$$

Further solving Eqs.(S37,S40), we obtain

$$\tau_s \frac{d\mathbf{s}}{dt} = \gamma\mathbf{u}^{-1}\left[\mathbf{\Lambda}\mathbf{s}^{\text{o}} - \left(\frac{\mathbf{u}}{\gamma}\mathbf{J}^{-1}\mathbf{G} + \mathbf{\Lambda}\right)\mathbf{s}\right] + m\mathbf{z}, \tag{S47}$$

$$\tau_z \frac{d\mathbf{z}}{dt} = -\mathbf{z} + \tau_z \frac{d\mathbf{s}}{dt} + \sqrt{\tau_z}\boldsymbol{\sigma}_z\boldsymbol{\xi}_1, \tag{S48}$$

where $\mathbf{u} = \text{diag}(u_1, ..., u_M)$, $\mathbf{J} = \text{diag}(J_1, ..., J_M)$, $\boldsymbol{\xi}_1 = \text{diag}(\xi_{1,1}, ..., \xi_{M,1})$, $\mathbf{G} = \{G_{ij}\}$ is a Laplacian matrix and $\boldsymbol{\sigma}_z\boldsymbol{\sigma}_z^T = 8a/(3\sqrt{3}\pi m)\sigma_V^2\mathbf{u}^{-1}$. The above dynamics correspond to Eq.(25-26) in the main text.

The above dynamics (Eq.(S47-S48)) implement HDF, and its stationary distribution equals to the target distribution, i.e., $\tilde{p}(\mathbf{s}) = p(\mathbf{s}|\mathbf{s}^{\text{o}})$. In particular, the stationary distribution of each feature sampled by each CANN equals to the corresponding marginal target distribution, i.e., $\tilde{p}(s_i) = p(s_i|\mathbf{s}^{\text{o}})$. This indicates that the coupled CANNs implement sampling-based Bayesian inference in a distributed way.

## 5 Sampling performances of coupled CANNs

We can re-organize Eq.(S47-S48) as:

$$\frac{d}{dt}\begin{pmatrix}\mathbf{s}\\\mathbf{z}\end{pmatrix} = -\begin{pmatrix}\tau_s^{-1}\boldsymbol{\alpha}^{-1}(\mathbf{L} + \mathbf{\Lambda}) & -m\tau_s^{-1}\boldsymbol{I}\\\tau_s^{-1}\boldsymbol{\alpha}^{-1}(\mathbf{L} + \mathbf{\Lambda}) & \tau_z^{-1} - m\tau_s^{-1}\boldsymbol{I}\end{pmatrix}\begin{pmatrix}\mathbf{s}\\\mathbf{z}\end{pmatrix} + \begin{pmatrix}\tau_s^{-1}\boldsymbol{\alpha}^{-1}\mathbf{\Lambda}\mathbf{s}^{\text{o}}\\\tau_s^{-1}\boldsymbol{\alpha}^{-1}\mathbf{\Lambda}\mathbf{s}^{\text{o}}\end{pmatrix} + \begin{pmatrix}0\\\boldsymbol{\sigma}_z/\sqrt{\tau_z}\boldsymbol{\xi}\end{pmatrix}, \tag{S49}$$

where $\boldsymbol{I}$ denotes the $M \times M$ identical matrix, $\mathbf{L} = \mathbf{u}\mathbf{J}^{-1}\mathbf{G}/\gamma$ and $\boldsymbol{\alpha} = \mathbf{u}/\gamma$.

We first solve the eigenvalues ($\lambda$) of the drift matrix, which satisfy,

$$\begin{vmatrix}\tau_s^{-1}\boldsymbol{\alpha}^{-1}(\mathbf{L} + \mathbf{\Lambda}) - \lambda\boldsymbol{I} & -m\tau_s^{-1}\boldsymbol{I}\\\tau_s^{-1}\boldsymbol{\alpha}^{-1}(\mathbf{L} + \mathbf{\Lambda}) & (\tau_z^{-1} - m\tau_s^{-1} - \lambda)\boldsymbol{I}\end{vmatrix} = 0. \tag{S50}$$

Note when $\lambda = \tau_z^{-1} - m\tau_s^{-1}$,

$$\begin{vmatrix}\tau_s^{-1}\boldsymbol{\alpha}^{-1}(\mathbf{L} + \mathbf{\Lambda}) - \lambda\boldsymbol{I} & -m\tau_s^{-1}\boldsymbol{I}\\\tau_s^{-1}\boldsymbol{\alpha}^{-1}(\mathbf{L} + \mathbf{\Lambda}) & 0\end{vmatrix} \neq 0. \tag{S51}$$

In the case $\lambda \neq \tau_z^{-1} - m\tau_s^{-1}$, Eq.(S50) becomes,

$$\left|\tau_s^{-1}\boldsymbol{\alpha}^{-1}(\mathbf{L} + \mathbf{\Lambda}) - \lambda\boldsymbol{I} + m\tau_s^{-1}(\tau_z^{-1} - m\tau_s^{-1} - \lambda)^{-1}\tau_s^{-1}\boldsymbol{\alpha}^{-1}(\mathbf{L} + \mathbf{\Lambda})\right| = 0. \tag{S52}$$

Denote the Jordan normal form of $\boldsymbol{\alpha}^{-1}(\mathbf{L} + \mathbf{\Lambda})$ is $\boldsymbol{\alpha}^{-1}(\mathbf{L} + \mathbf{\Lambda}) = \boldsymbol{P}\boldsymbol{Q}\boldsymbol{P^T}$. Eq.(S50) is written as

$$\left|\tau_s^{-1}\boldsymbol{Q} - \lambda\boldsymbol{I} + m\tau_s^{-1}(\tau_z^{-1} - m\tau_s^{-1} - x)^{-1}\tau_s^{-1}\boldsymbol{Q}\right| = 0. \tag{S53}$$

Denote $i$-th diagonal element of the matrix $\boldsymbol{Q}$ as $Q_i$, and rank them in the descending order, i.e., $Q_i > Q_j$, for $i < j$. Since $\boldsymbol{\alpha}^{-1}(\mathbf{L} + \mathbf{\Lambda})$ is a general symmetric matrix, all the eigenvalues are real numbers. Eq.(S50) is equivalent to,

$$\tau_s^{-1}Q_i - \lambda + m\tau_s^{-1}(\tau_z^{-1} - m\tau_s^{-1} - \lambda)^{-1}\tau_s^{-1}Q_i = 0, \quad i = 1, ..., M. \tag{S54}$$

Solving the above equation, we obtain,

$$\lambda_i^{\pm} = \frac{1}{2}\left(-\tau_s^{-1}m + \tau_z^{-1} + \tau_s^{-1}Q_i \pm \sqrt{(-\tau_s^{-1}m + \tau_z^{-1} + \tau_s^{-1}Q_i)^2 - 4\tau_z^{-1}\tau_s^{-1}Q_i}\right). \tag{S55}$$

The real-part of the smallest eigenvalue $h$ determines the convergence of the dynamic system. It can be checked that $\mathrm{Re}(\lambda_M^-) \leq \mathrm{Re}(\lambda_i^-)$ and $\mathrm{Re}(\lambda_M^+) \leq \mathrm{Re}(\lambda_i^+)$, for $i < M$. Thus, $h = \min\left(\mathrm{Re}(\lambda_M^-), \mathrm{Re}(\lambda_M^+)\right) = \mathrm{Re}(\lambda_M^-)$ corresponding to Eq.(30) in main text.

It is straightforward to check that:

- When $0 < m \leq m_{max} = (\sqrt{\tau_s/\tau_z} - \sqrt{Q_M})^2$, $h$ monotonically increases with $m$ and $h > 0$.

- When $m > m_{max}$, $h$ monotonically decreases with $m$. In particular, when $m_{max} < m < m_{th} = \tau_s/\tau_z + Q_M$, $h > 0$; when $m > m_{th}$, $h < 0$ indicating the divergence of the dynamic system.

Thus, the coupled CANNs performs HDF when $0 < m < m_{th}$. In particular, when $m = m_{max}$, $h$ reaches the maximum value, and the sampling reaches to the fastest speed.