# OpenReview forum: "Adaptation Accelerating Sampling-based Bayesian Inference in Attractor Neural Networks"
_NeurIPS.cc/2022/Conference — NeurIPS 2022 Accept_

### Official Review · Reviewer_3aPt · 2022-06-17

**Rating:** 7
**Confidence:** 4
**Soundness:** 3 good
**Presentation:** 3 good
**Contribution:** 3 good

**Summary:**

The paper shows that a particular model of continuous attractors very relevant in neuroscience performs HMC sampling.

**Questions:**

N/A

**Limitations:**

It isn't clear how far the basic ideas here will extend (e.g. to learned representations, visual system etc.)  But it is definitely an approach worth considering in future.

**Strengths And Weaknesses:**

Strengths:
* It is actually surprisingly hard to make rigorous connections between HMC sampling and real neural circuits.  In the Aitchison and Lengyel, they were for instance forced to use linear networks, without the spiking nonlinearity.  But they managed to make such a connection here!  The key innovation here was to consider HMC sampling over higher-level variables such as the location of the bump of activity, rather than considering e.g. firing rates themselves as the variables that HMC samples.

The key weakness is the need for clarification of the actual derivation of the equivalence, which is currently quite unclear.  Here are some specific issues (but my comments are not limited to these issues):
* Eq. 15: no x on RHS.
* Eq. 15/16 G() vs G[] (i.e. sometimes G has curly, sometimes square brackets)
* The projection seems like the key step.  Can we give some flavour of how that works in the main text, and whether any assumptions go in there (e.g. linearity).
* It would be worth exploring the biology.  Is there really this kind of adaptation present in e.g. head-direction cells?  That would seem strange, as it would destabilise the bump.  This mechanism would be a potential explanation.

---

> ### Author Response · Authors · 2022-08-01
> **Reply to Reviewer 3aPt**
>
> We acknowledge the encouraging and insightful comments of the reviewer, which concisely summarize the contributions of our work. In the below, we would like to address the questions of the reviewer.
>
> **Eq. 15: no $x$ on RHS and Eq. 15/16 $G(\cdot)$ vs $G[\cdot]$**
>
> Thanks for pointing out these mistakes. We will correct them in the revised manuscript.
>
> **The projection seems like the key step. Can we give some flavour of how that works in the main text, and whether any assumptions go in there (e.g. linearity).**
>
> Yes, the projection method is the key strategy in our theoretical analysis. This method comes from the unique property of CANNs, that is, due to the translation-invariant recurrent connections between neurons, the dynamics of a CANN is dominated by very few motion modes (the eigenvalues of motion modes decay exponentially), and the first two dominating motion modes are the position and height changes of the bump. Utilizing this property, we can then use the projection method to reduce the dimension of the network dynamics significantly, and the reduced dynamics allows us analyze the model behaviors theoretically. The rigorous analysis of the projection method can be found in [1], and the use of the projection method to simplify the model dynamics is given in SI. As suggested by the reviewer, we will add more descriptions about the project method in the revised manuscript.
>
> *Reference:*
>
> [1] Fung, CC Alan, KY Michael Wong, and Si Wu. "A moving bump in a continuous manifold: a comprehensive study of the tracking dynamics of continuous attractor neural networks." Neural Computation 22.3 (2010): 752-792.
>
> **It would be worth exploring the biology. Is there really this kind of adaptation present in e.g. head-direction cells? That would seem strange, as it would destabilize the bump. This mechanism would be a potential explanation.**
>
> Adaptation is a very general property of neural systems widely found in the brain [1]. A number of mechanisms can lead to spike-frequency adaptation (SFA). At the single neuron level, there are three mechanisms: 1) the current caused by voltage-dependent, high-threshold potassium channels; 2) the current mediated by calcium-dependent potassium channels; 3) the current caused by slow recovery from in-activation of fast sodium channel. In brain areas where the CANN is involved, adaptation (SFA) was reported by experiments, e.g. in hippocampus [2] and in visual cortex [3].
>
> *Reference:*
>
> [1] Ha, Go Eun, and Eunji Cheong. "Spike frequency adaptation in neurons of the central nervous system." Experimental neurobiology 26.4 (2017): 179.
>
> [2] Gu, Ning, Koen Vervaeke, and Johan F. Storm. "BK potassium channels facilitate high‐frequency firing and cause early spike frequency adaptation in rat CA1 hippocampal pyramidal cells." The Journal of physiology 580.3 (2007): 859-882.
>
> [3] Sanchez-Vives, Maria V., Lionel G. Nowak, and David A. McCormick. "Cellular mechanisms of long-lasting adaptation in visual cortical neurons in vitro." Journal of Neuroscience 20.11 (2000): 4286-4299.
>
> **We hope that our replies have addressed the concerns of the review.**

---

### Official Review · Reviewer_mjGP · 2022-07-06

**Rating:** 5
**Confidence:** 3
**Soundness:** 3 good
**Presentation:** 2 fair
**Contribution:** 2 fair

**Summary:**

In this manuscript the authors describe how adaptation can speed up posterior sampling in continuous attractor neural networks, a neural population model known to be capable of posterior sampling. Concretely, they show that the dynamics of networks with adaptation is equivalent to Hamiltonian dynamics with friction. This equivalence gives a theoretical justification why the methods with adaptation perform better than the ones without adaptation, which effectively implement (first order) Langevin dynamics. Furthermore, their theoretical analysis allows the authors to compute good parameter settings to sample from simple one and two dimensional Gaussians, which are confirmed in simulations.

**Questions:**

If the authors can show robustness in some way, i.e. that this works for other distributions, this would go a long way towards convincing me that their sampling method could work for other distributions where sampling is necessary.

**Limitations:**

---

**Strengths And Weaknesses:**


The observation that an adaptation mechanism that pushes the sampling chain away from previously visited states can improve convergence speed makes intuitive sense and the derivations in the paper seem correct. Also, the formal connection to hamiltonian sampling with friction are interesting. However, parts of the theory and all of the evaluation are specific to Gaussian probability distributions, for which sampling is not strictly necessary. Thus, this paper is an interesting step towards understanding the dynamics of sampling networks, but not a big break through.

Concrete points:
1) I am generally worried that all theory here is implemented only for Gaussians and some of the parameters like the optimal and threshold adaptation strength depend on the input distribution. Computing with Gaussians really does not require sampling or a whole population of neurons arranged along the feature space. To make sampling sensible at least arbitrary distributions should be covered I think.
2) The setup for the CANNs discussed here seems a bit odd, because the input current in itself already a representation of the distribution, which allows computation as discussed in the probabilistic population coding framework. It is unclear to me why one would want to sample from a distribution given in this format.
3) Without a deep dive into the CANN literature, some of the derivations are extremely hard to follow. In particular, which properties are first derived here, which ones are taken from which older literature and under which assumptions these results hold is impossible to take from the manuscript alone. Here, the authors could definitely communicate more clearly.


Small points:
In (14) the standard deviation in the exponential is a, but should be Lambda.

---

> ### Author Response · Authors · 2022-08-01
> **Reply to Reviewer mjGP (part 2)**
>
> **On the external input Eq. 14.**
>
> The external input in Eq.14, $I(x)=\Lambda \exp[-(x-s^\rm{o})^2/(4a^2)]$, can actually be regarded as the neuronal responses in the sensory layer which convey the stimulus information to the CANN, where the neuronal tuning function has the Gaussian shape and $x$ is the preferred stimulus value of the neuron. Note that the standard deviation in the exponential is the tuning width $a$, not the peak firing rate $\Lambda$. In the framework of PPC, the inverse of $\Lambda$ is interpreted as the uncertainty of the stimulus information.
>
> Notably, in the high-dimensional case of our model (coupled-CANNs), the external input conveys the likelihood distribution, while the prior distribution of stimulus features is stored in the reciprocal connections between CANNs (Eq.29), and no posterior distribution is given. It is the dynamics of couple-CANNs “computing” the posterior and performing sampling in the posterior space (see lines274-280).
>
> **On the clarity of theoretical derivations**
>
> The projection method and some fundamental properties of CANNs were introduced in previous works. Here, our new contributions are on applying the projection method to formally derive how the CANN dynamics with adaptation implements the standard HMC and analyzing the sampling behaviors of the network systematically. As suggested by the reviewer, we will add more descriptions in the revised manuscript to make the derivations clearly.
>
> **We hope that our replies have clarified the concerns of the reviewer.**

---

> ### Author Response · Authors · 2022-08-01
> **Reply to Reviewer mjGP (part 1)**
>
> We acknowledge the valuable comments of the reviewer and would like to clarify the concerns of the reviewer in the below.
>
> **On sampling Gaussian vs. arbitrary distributions**
>
> 1. It is still an open question whether the brain samples arbitrary complex distributions, and if so, when and where the brain does this. There are reports that the brain perceives complex multimodal distributions, while there are also reports that people tend to mistakenly recall complex multimodal distribution to be unimodal [1]. Interestingly, in visual perception where the CANN is widely used as a computational model (see, e.g., [2]), it was reported that people tend to ignore higher-order statistical moments of inputs [3]. It seems that different brain areas have different requirements on representing distributions, and in some areas (such as the visual system), sampling a Gaussian distribution may already be adequate, as suggested by [3].
> 2. It is worth to mention that even for the Gaussian distribution, sampling stimulus values is still necessary in the brain. It has remained debated in the field as how the brain represents the uncertainty of information (often consider the Gaussian distribution as the example). The theory of probabilistic population code (PPC) assumes that the brain extracts the mean and variance of the Gaussian distribution explicitly by utilizing the Poisson statistics of neuronal responses, but exactly how the brain reads out this information remains unclear. The sampling-based theory assumes that the neural system reads out the stimulus information through sampling the posterior distribution, which is consistent with the irregular firings of neurons. Here, we adopt the sampling point of view.
> 3. In the current study, we consider that the neuronal recurrent connection profile is Gaussian, which leads to that the network dynamics samples the Gaussian posterior. Alternatively, if we set the neuronal connection profile to be the Von Mises distribution, then the network dynamics samples the Von Mises posterior. In theory, if we can set the neuronal recurrent connections properly, the network can sample an arbitrarily given distribution. However, a single CANN with a fixed connection pattern can only sample a fixed form of distribution. Applying a single neural circuit to sample many distributions may be computationally interesting, but may not be biologically relevant, since the brain can easily employ different neural circuits to sample different distributions.
> 4. In case that the CANN samples the Gaussian posterior, while the true posterior is non-Gaussian, then the network performs a Gaussian approximation, similar to the idea of Variational Inference in machine learning. This mismatching may explain the experimental findings in [3], and we will explore this issue in the future work (thank the reviewer for raising this interesting question).
> 5. Overall, the current study has focused on studying the case of sampling the Gaussian posterior (and so did most previous sampling works in computational neuroscience, see, e.g., [4-5]), and the simple model has the advantage of allowing us to analytically elucidate how the neural circuit realizes the optimal Hamiltonian Monte Carlo sampling. Nevertheless, since the CANN with Gaussian tuning is a canonical network model in neural systems, our study already has potentials to explain the sampling of a number of stimulus features, including orientation, head-direction, and spatial location etc. Thus, we believe that our work is an important step towards understanding how sampling inference is accelerated in the brain.
>
> *References:*
>
> [1] Nisbett, Richard E., and Ziva Kunda. "Perception of social distributions." Journal of Personality and Social Psychology 48.2 (1985): 297.
>
> [2] Ben-Yishai, Rani, R. Lev Bar-Or, and Haim Sompolinsky. "Theory of orientation tuning in visual cortex." Proceedings of the National Academy of Sciences 92, no. 9 (1995): 3844-3848.
>
> [3] Waskom, M. L., J. Asfour, and R. Kiani. "Perceptual insensitivity to higher-order statistical moments of coherent." (2018).
>
> [4] Echeveste, Rodrigo, et al. "Cortical-like dynamics in recurrent circuits optimized for sampling-based probabilistic inference." Nature neuroscience 23.9 (2020): 1138-1149.
>
> [5] Savin, Cristina, and Sophie Denève. "Spatio-temporal representations of uncertainty in spiking neural networks." Advances in Neural Information Processing Systems 27 (2014).

---

> ### Author Response · Authors · 2022-08-09
> **Followup comments**
>
> Thank you for your questions/comments and your time and effort in reviewing the paper. Just wondering if our comments resolved your question/concerns.  If this is not the case, we hope the reviewer could clarify any remaining concerns/questions and we are happy to engage in further discussion.

---

### Official Review · Reviewer_YNCK · 2022-07-11

**Rating:** 7
**Confidence:** 4
**Soundness:** 3 good
**Presentation:** 3 good
**Contribution:** 3 good

**Summary:**

In this paper, the authors address the question of how sampling-based probabilistic inference could be implemented in the brain. An important concern about sampling-based theories is that the sampling speed is too slow to be compatible with brain function.

One potential approach that circumvents the speed issue is Hamiltonian Monte Carlo (HMC) sampling. Previous works have shown that HMC can be implemented by a biologically plausible neural network with balanced excitation-inhibition among neurons.
Here, they present an alternate neural implementation of HMC using a canonical network model called continuous attractor neural networks (CANNs).

Using a linear Gaussian generative model, the authors analytically show how the dynamics of a CANN are equivalent to HMC. Further, they show that adding a noisy adaptation term in the CANN dynamics accelerates the sampling process. They also show how this model could be extended to sampling from a multivariate Gaussian posterior. Finally, they present simulation results that confirm their theoretical analyses.

**Questions:**

- There are certain assumptions placed on the form of the neuronal connection strengths, firing rate, adaptation, and external input for the CANN dynamics to be equivalent to HMC sampling for a linear Gaussian model. Are all of these biologically plausible? How would the dynamics change if any of these assumptions are relaxed?
- Are there any thoughts about how this approach could be extended to multimodal, non-Gaussian distributions?
- The speed-up observed here is relative to first-order Langevin dynamics. Could you provide a sense of how this compares with inference time scales observed in neuroscience experiments?
- Clarifying question: in eq 2, where the prior is assumed to be uniform, is the support of $s$ finite?
- Are there any concrete experimental predictions that could arise from this work?


**Limitations:**

There is no potential negative societal impact - the authors state that their study is fundamental in neuroscience and not tied to applications.

The authors also discuss the limitations of their work - the analytical results derived here and the simulations correspond to a linear Gaussian generative model. It would be interesting to see how their work could be extended to performing inference on more complicated, multimodal distributions.

**Strengths And Weaknesses:**

This work demonstrates how continuous attractor neural networks (CANNs) could implement Hamiltonian Monte Carlo (HMC) sampling. The authors show that (i) when the dynamics of the adaptation current in CANNs take the form of spike frequency adaptation, and (ii) CANN dynamics are projected onto the first two dominating motion modes, the resulting dynamics of the latent features are equivalent to HMC.
In figure 2 they also clearly show how varying the adaption strength results in different sampling regimes. In particular, the simulation results show how the sampling is sped up as the adaptation strength increases in the HMC regime. Overall, this is an interesting implementation of HMC with friction using CANNs.

As the authors mention in the discussion, the results presented here are for a linear Gaussian model. Real-world distributions are more complicated and often multimodal. It would be interesting to consider how the current approach could be extended to perform inference on more complicated distributions.
It would also be useful to provide a sense of how the speedup obtained here compares with inference timescales observed in the brain.

The paper is quite clear and the figures are very neat and well-annotated.
However, there are a few minor typos and grammatical errors.

---

> ### Author Response · Authors · 2022-08-01
> **Reply to Reviewer YNCK (part 2)**
>
> **Is the support of $s$ finite?**
>
> In theoretical analysis, we consider the support of $s$ is infinite ($s \in \mathbb{R}$). For some stimulus features, such as orientation, $s \in (-\pi, \pi]$ with a periodic boundary. Consider the neural interaction range $a$ is much smaller than $2\pi$ in our model, treating the support of $s$ to be infinite is still a good approximation.
>
> **On the experimental predictions**
>
> Our model can generate testable experimental predictions. Below lists two of them.
>
> 1. Our model predicts that reaction times of neurons are positively correlated with their oscillation frequencies in sampling-based Bayesian inference. According to our theoretical analysis, when the SFA strength $m$ is larger than the critical value for realizing the maximum sampling speed (this always occurs due to adaptation noise), the eigenvalues of the drift matrix become complex numbers (i.e., they have imaginary parts). This leads to oscillation of neurons in the network dynamics, and the sampling speed decreases the oscillation frequency. Thus, in a task involving Bayesian inference, e.g., probabilistic decision making, we may record the oscillation frequencies of neurons and their reaction times. Our model predicts that the reaction time is positively correlated with the oscillation frequency.
> 2. Our model predicts that the sampling speed increases with the SFA strength $m$ for $m<m_{max}$, where $m_{max}$ is the critical value for achieving the maximum sampling speed. Suppose one can manipulate the SFA strength in the experiment, then we can test this prediction. There are a number of mechanisms responsible for SFA, and three of them are: 1) the current caused by voltage-dependent, high-threshold potassium channels; 2) the current mediated by calcium-dependent potassium channels; 3) the current caused by slow recovery from in-activation of fast sodium channels. Consider the calcium-dependent potassium channels as an example. We can perform whole cell recording of neurons in a brain slice in vitro to test whether calcium-gated potassium channels could reduce the effect of SFA by applying a channel blocker, e.g., Iberiotoxin K_{Ca} (BK channel blocker), Apamin K_{Ca^2} (SK channel blocker), Lei-Dab 7 (high affinity and selective K_{Ca^{2.2}} (SK) blocker). If indeed the SFA effect can be controlled by applying ion channel blockers, we can then test its effect on the sampling speed as predicted by our model. If indeed the SFA effect can be controlled by applying ion channel blockers, we can then test its effect on the sampling speed as predicted by our model.
>
> **We hope that our replies have addressed all concerns of the reviewer.**

---

> ### Author Response · Authors · 2022-08-01
> **Reply to Reviewer YNCK (part 1)**
>
> We acknowledge the encouraging and insightful comments of the reviewer, and would like to address the questions of the reviewer in details below.
>
> **On the generality of the results**
>
> To pursue theoretical analysis, we have considered a canonical network CANN, a simple form of adaptation dynamics, and an external input in the form of Gaussian tuning function. This simple model allows us to solve the network dynamics analytically and elucidate the sampling acceleration mechanism clearly. All these simplifications are believed to be biologically plausible, as they are widely used in modelling studies in the field. Relaxing these assumptions to a large extent will not change our main results: as long as the network can hold a family of attractors and that the adaptation can destabilize these attractor states, then the network can perform the HMC dynamics approximately.
>
> **On the extension to multimodal, non-Gaussian distributions**
>
> Thank the reviewer for raising this important question.
> 1. On the computational theory perspective, it is still an open question whether the brain processes complicated distributions, and if so, when and where the brain does this. There are reports that the brain perceives multimodal distributions, while there are also reports that people tend to mistakenly recall multimodal distribution to be unimodal [1]. Interestingly, in visual perception where the CANN is widely used as a computational model (see, e.g., [2]), it was reported that people tend to ignore higher-order statistical moments of inputs [3]. It seems that different brain areas have different requirements on representing distributions, and in some areas (such as the visual system), sampling a Gaussian distribution may already be adequate.
> 2. For our network model, a single CANN can only sample a unimodal distribution, and the form of distribution depends on the profile of neuronal recurrent connections. For instance, in the current study, we set the neuronal connectional profile to be Gaussian, then the network dynamics samples the Gaussian posterior; if we set the neuronal connection profile to be the Von Mises distribution, then the network dynamics samples the Von Mises posterior.
> 3. We did observe that in coupled-CANNs, when the range of reciprocal connections between CANNs is much smaller than the range of neuronal recurrent connections in individual CANNs, the network dynamics can sample a multimodal distribution. However, we have not investigated this issue carefully, so we would prefer to report the result in the future work.
> 4. Overall, our work model can implement the sampling dynamics for a unimodal distribution, and particularly, the Gaussian posterior. This may appear to a limitation in computation, however, consider that we are studying brain functions and that the CANN is a canonical network in the brain, our model can already explain the sampling of a number of stimulus features, including orientation, head-direction, and spatial location.
>
> *References:*
>
> [1] Nisbett, Richard E., and Ziva Kunda. "Perception of social distributions." Journal of Personality and Social Psychology 48, no. 2 (1985): 297.
>
> [2] Ben-Yishai, Rani, R. Lev Bar-Or, and Haim Sompolinsky. "Theory of orientation tuning in visual cortex." Proceedings of the National Academy of Sciences 92, no. 9 (1995): 3844-3848.
>
> [3] Waskom, Michael L., Janeen Asfour, and Roozbeh Kiani. "Perceptual insensitivity to higher-order statistical moments of coherent random dot motion." Journal of vision 18, no. 6 (2018): 9-9.
>
> **On the sampling speed**
>
> In the neuroscience society, it remains unknown yet the exact time cost for the brain performing Bayesian inference. The only known fact is that this process is extremely fast [1]. In monkey experiments, the electrophysiology studies revealed that the visual cortex can accomplish a probabilistic decision-making task in less than 800ms [2] and a contour integration task in less than 200ms [3]. If we plug in the biological relevant parameter value $\tau_s=$5-10ms in the model, it gives that the sampling speed of HDF is around 100-200ms, while the speed of FLD is around 2-4s in a single CANN (see Fig.2B-C). Hence, HDF is about 20 times faster than FLD.
>
> *References:*
>
> [1] Knill, David C., and Whitman Richards, eds. Perception as Bayesian inference. Cambridge University Press, 1996.
>
> [2] Nienborg, Hendrikje, and Bruce G. Cumming. "Decision-related activity in sensory neurons reflects more than a neuron’s causal effect." Nature 459, no. 7243 (2009): 89-92.
>
> [3] Chen, Minggui, Yin Yan, Xiajing Gong, Charles D. Gilbert, Hualou Liang, and Wu Li. "Incremental integration of global contours through interplay between visual cortical areas." Neuron 82, no. 3 (2014): 682-694.

---

> ### Author Response · Authors · 2022-08-09
> **Followup comments--any feedback?**
>
> We hope our replies and corresponding changes in the updated paper have addressed all concerns/questions raised by the reviewer. If this is not the case, we hope the reviewer could clarify any remaining concerns/questions and we are happy to engage in further discussion.
>
> We again thank the reviewer for the thoughtful comments and for the time and effort in reviewing the paper.

---

> ### Comment · Reviewer_YNCK · 2022-08-09
> **Response to author rebuttal**
>
> Thank you for the detailed responses to my questions. After taking these into consideration, I have increased my rating by 1 point.
> It would be useful to include some points on the extensions to multimodal distributions and experimental predictions in the discussion.

---

> > ### Author Response · Authors · 2022-08-09
> > **Response to the comment**
> >
> > We thank the reviewer for the comments. Yes, we will add the points on the extensions to multimodal distributions and experimental predictions in the revised manuscript.

---

### Official Review · Reviewer_trHg · 2022-07-15

**Rating:** 7
**Confidence:** 2
**Soundness:** 3 good
**Presentation:** 4 excellent
**Contribution:** 3 good

**Summary:**

In this work, the authors:
- review Bayesian inference for linear gaussian models, as well as HMC
- derive how the equations of a CANN with noisy adaptation is identical to that of HMC, with the appropriate substitutions, and how parameter regimes translates to sampling behavior
- demonstrate how CANN with noisy adaptation converges to the true posterior (in mean, variance, and DKL) more quickly than without, in single and coupled networks

**Questions:**

- the work is motivated by the claim that naive Langevin sampling cannot work at perceptually relevant speeds due to its slow dynamics around the posterior mode. This is not strictly true, since speed can come at a cost of certainty, and it's unclear if FLD performs unreasonably (and unrealistically) poorly compared to human performance. While it's convincingly shown that higher adaptation (up to the derived max speed) increases sampling efficiency, I'm curious how the performance of both algorithms are when measured against absolute time, instead of in units of synaptic time constant tau_s (e.g., in Figure 2B). In other words, if one plugs in a reasonable value for tau_s (e.g., 5-10ms for AMPA receptors), would HDF actually achieve reasonable performance in reasonable time?

- a critical component of the model seems to be the stochasticity in the adaptation dynamics (Eq13). To my limited knowledge, this is not a common assumption for CANNs, and is typically not included for adaptation dynamics in other models (such as spiking neuron). I would ask the authors to provide some evidence for why this assumption is justified (other than the currently cited "biological systems is noisy"). While SI3 provides some evidence that small deviations from the exact value of sigma_V is fine, perhaps a broader treatment is necessary, since one typically assumes the cell has near-perfect access to its own spike history (through calcium dynamics, etc)

- my only concern wrt to originality / significance is, how different is this work from ref 16 & 18? In other words, does CANN need an entirely different treatment than from spiking networks, which is arguably a more difficult problem? This question stems from my naivety, but it would be good if the authors further discuss why CANN should be significantly different, perhaps from the perspective of adaptation (or something else)

**Limitations:**

very brief discussion of limitation is included, and while the robustness of model parameters (in relation to deriving an exact match to HMC) is discussed in the SI, it's worth mentioning again in the limitations, as well as other strong assumptions of the model. No discussion of societal impact (and none from my perspective)

**Strengths And Weaknesses:**

originality: spiking networks have been shown to approximate HMC sampling (in ref 16 & 18). The current authors extend this investigation  to rate-based bump attractor networks, which is commonly used in computational neuroscience. I don't know of other works on this particular topic of adaptation, though I'm not well read in this literature (but see https://www.biorxiv.org/content/10.1101/2021.03.15.434192v2)

quality: at a superficial level and without re-deriving the proofs, I believe the work is of high quality, and that the results fully support the central claim of efficiency (but not necessarily the claim of biologically relevant speed, expanded below)

clarity: overall, I find this paper to be of high quality, and very clearly presented, even for a relative outsider of the literature. The brief but concise review of Bayesian inference, HMC and CANNs touched on all the relevant background without unnecessary length, and I commend the authors for writing such an enjoyable paper and without assuming basic knowledge from the reader, since I learned much from reading it. The figures are also well-presented and with readable font sizes.

significance: I believe this paper links two very interesting ideas: adaptation and neural implementation of probabilistic sampling, and will be of significance for a segment of the computational neuroscience community

---

> ### Author Response · Authors · 2022-08-01
> **Reply to Reviewer trHg (part 2)**
>
> **On the difference to Ref. 16 & 18**
>
> There are a number significant differences between our work and Ref.16,18, which are:
> 1. The encoding paradigms are different. In Ref.16,18, the authors consider that the sampled features are encoded by individual neurons, while we consider that the sampled feature is encoded by neural population (i.e., the CANN). See descriptions in lines 46&154 in the paper.
> 2. The models used are different. Ref.16,18 consider linear networks, while we include nonlinearity in the model (Eq.12).
> 3. The details of the sampling dynamics are different. In Ref.16,18, to inject noises, the authors use a modified HMC with mixed first-order Langevin dynamics, while our model realizes the standard HDF sampling dynamics.
> 4. The momentum terms in HMC are different. In Ref.16,18, the authors consider that the dynamics of a group of inhibition neurons serve as the momentum term, while in our model, SFA effectively induces the momentum effect in HMC.
>
> **Limitation**
>
> As suggested by the reviewer, we will add more discussions about the limitation and restrictions of our model in the revised manuscript.
>
> **On the biorxiv paper**
>
> Since the reviewer mentioned a biorxiv paper in the comments,
> (https://www.biorxiv.org/content/10.1101/2021.03.15.434192v2)
> we read this paper and confirmed that it is about evidence accumulation in decision-making and is not relevant to our study. To our best knowledge, we have included all references on sampling acceleration and also compared them to our model.
>
> **We hope that we have clarified all concerns of the reviewer.**

---

> > ### Comment · Reviewer_trHg · 2022-08-08
> > **thanks for the response**
> >
> > thanks to the authors for answering my questions: I think these points regarding the assumptions and limitations of the model in the context of the biological brain are important to mention in the discussion, and I am satisfied with their response otherwise.

---

> > > ### Author Response · Authors · 2022-08-09
> > > **Thanks for the comments**
> > >
> > > We thank the reviewer for the comments. Yes, we will add the assumptions and limitations of the model in the revised manuscript.

---

> ### Author Response · Authors · 2022-08-01
> **Reply to Reviewer trHg (part 1)**
>
> We acknowledge the encouraging and valuable comments of the reviewer, and would like to address the questions of the reviewer in details below.
>
> **On the sampling speed**
>
> 1. As suggested by the reviewer, we plug in the biological relevant parameter value $\tau_s=$5-10ms, and obtain that the sampling speed of HDF is around 100-200ms, while the speed of FLD is around 2-4s in a single CANN (see Fig.2B-C). This gives that HDF is about 20 times faster than FLD in the 1D feature case. Notably, for sampling high-dimensional stimulus features (the case of coupled CANNs), the speed of FLD is further delayed compared to HDF (see Fig.3C, N=5).
>
> 2. To our best knowledge, we could not find a formal report about the time cost of perceptual inference in the brain, but a large volume studies has indicated that this is an extremely fast process, see, e.g., the studies in the book [1]. In human experiments, the reaction time of subjects in a perceptual inference task is typically around 1-2s [2,3], which includes the extra time for processing sensory inputs and accomplishing motor behaviors. In monkey experiments, the electrophysiology studies suggest that the visual cortex can perform a decision-making task in less than 800ms (Fig.2c in [4]) and a contour integration task in less than 200ms (Fig.3 in [5]).
>
> 3. Overall, the experimental evidence suggest that sampling acceleration is needed, if the brain does employ the sampling-based Bayesian inference.
>
> *References:*
>
> [1] Knill, David C., and Whitman Richards, eds. Perception as Bayesian inference. Cambridge University Press, 1996.
>
> [2] Fetsch, Christopher R., Gregory C. DeAngelis, and Dora E. Angelaki. "Bridging the gap between theories of sensory cue integration and the physiology of multisensory neurons." Nature Reviews Neuroscience 14, no. 6 (2013): 429-442.
>
> [3] Laquitaine, Steeve, and Justin L. Gardner. "A switching observer for human perceptual estimation." Neuron 97, no. 2 (2018): 462-474.
>
> [4] Nienborg, Hendrikje, and Bruce G. Cumming. "Decision-related activity in sensory neurons reflects more than a neuron’s causal effect." Nature 459, no. 7243 (2009): 89-92.
>
> [5] Chen, Minggui, Yin Yan, Xiajing Gong, Charles D. Gilbert, Hualou Liang, and Wu Li. "Incremental integration of global contours through interplay between visual cortical areas." Neuron 82, no. 3 (2014): 682-694.
>
> **On the stochasticity of the adaptation dynamics**
>
> In our model, we consider spike frequency adaptation (SFA) to implement adaptation. A number of mechanisms in biological systems can realize SFA, and three of them are often studied [1], which are: 1) the current caused by voltage-dependent, high-threshold potassium channels; 2) the current mediated by calcium-dependent potassium channels; 3) the current caused by the slow recovery from in-activation of fast sodium channels. All these mechanisms depend on ion concentrations, release of neural transmitters, activation/inactivation of ion channels, buffering and diffusion, and all these processes are very noisy (see Fig.9 in [2]). Recent works also show that adaptation noises can play important computational roles (see, e.g., [3]). Indeed, previous modeling works rarely consider adaptation noises, since they focused on different things rather than the functions of adaptation noises. Here, we show that adaption noises can actually contribute to accelerate stochastic sampling.
>
> As suggested by the reviewer, we will add discussions about the biological plausibility of adaptation noise in the revised paper.
>
> *References:*
>
> [1] Benda, Jan, and Andreas VM Herz. "A universal model for spike-frequency adaptation." Neural computation 15, no. 11 (2003): 2523-2564.
>
> [2] Alonso, Angel, and Ruby Klink. "Differential electroresponsiveness of stellate and pyramidal-like cells of medial entorhinal cortex layer II." Journal of neurophysiology 70, no. 1 (1993): 128-143.
>
> [3] Fisch, Karin, Tilo Schwalger, Benjamin Lindner, Andreas VM Herz, and Jan Benda. "Channel noise from both slow adaptation currents and fast currents is required to explain spike-response variability in a sensory neuron." Journal of Neuroscience 32, no. 48 (2012): 17332-17344.

---

### Meta-Review · Area_Chair_2nMt · 2022-08-26

**Recommendation:** Accept
**Confidence:** Certain

**Metareview:**

The reviewers agree that the paper makes an interesting contribution, connecting inference in probabilistic models with network models from computation neuroscience.

**Award:**

No

---

### Decision · Program_Chairs · 2022-09-14

Accept